# Hydrological Response of Andean Catchments to Recent Glacier Mass Loss

Alexis Caro[1], Thomas Condom[1], Antoine Rabatel[1], Nicolas Champollion[1], Nicolás García[2], Freddy Saavedra[3]

[1] Univ. Grenoble Alpes, CNRS, IRD, INRAE, Grenoble-INP, Institut des Géosciences de l'Environnement, Grenoble, France

[2] Glaciología y Cambio Climático, Centro de Estudios Científicos (CECs), Valdivia, Chile

[3] Departamento de Ciencias Geográficas, Facultad de Ciencias Naturales y Exactas, Universidad de Playa Ancha, Leopoldo Carvallo 270, Playa Ancha, Valparaíso, Chile

*Correspondence to:* Alexis Caro (alexis.caro.paredes@gmail.com)

**Abstract.** The impacts of the accelerated glacier retreat in recent decades on glacier runoff changes are still unknown in most Andean catchments, increasing uncertainties in estimating water availability. This particularly affects the Outer Tropics and Dry Andes, heavily impacted by prolonged droughts. Current global estimates overlook climatic and morphometric disparities among Andean glaciers, which significantly influence model parameters. Meanwhile, local studies have used different approaches to estimate glacier runoff in a few catchments. Improving 21st-century glacier runoff projections relies on calibrating and validating models using corrected historical climate inputs and calibrated parameters across diverse glaciological zones. Here, we simulate glacier evolution and related runoff changes between the periods 2000-2009 and 2010-2019 across 786 Andean catchments (11,282 km$^2$ of glacierized area, 11°N-55°S) using the Open Global Glacier Model (OGGM). TerraClimate atmospheric variables were corrected using *in situ* data, getting a mean temperature bias by up to 2.1°C and enhanced monthly precipitation. Glacier mass balance and volume were calibrated, where melt factor and Glen A parameter exhibited significant alignment with varying environmental conditions. Simulation outcomes were validated against *in situ* data in three documented catchments (with a glacierized area > 8%) and monitored glaciers. Our results at the Andes scale reveal an average reduction of 8.3% in glacier volume and a decrease of 2.2% in surface area between the periods 2000-2009 and 2010-2019. Comparing these two periods, glacier and climate variations have led to a 12% increase in mean annual glacier melt (86.5 m$^3$/s) and a decrease in rainfall on glaciers of -2% (-7.6 m$^3$/s) across the Andes, both variables compose the glacier runoff. We confirmed the utility of our corrected regional simulations of glacier runoff contribution at the catchment scale, where our estimations align with previous studies (*e.g.*, Maipo 34°S, Chile), provide new insights on the seasonal glaciers' largest contribution (*e.g.*, La Paz 16°S, Bolivia) and new estimates of glacier runoff contribution (*e.g.*, Baker 47°S, Chile).

## 1 Introduction

The largest glacierized area in the southern hemisphere outside the Antarctic ice sheet is found in the Andes (RGI Consortium, 2017; Masiokas et al., 2020). Andean glaciers supply water for roughly 45% of the population in the Andean countries (Devenish and Gianella, 2012) and for ecosystems (Zimmer et al., 2018; Cauvy-Fraunié

and Dangles, 2019). Continuous glacier shrinkage has been detected since the late 1970s, with intensification observed over the past two decades (Rabatel et al., 2013; Dussaillant et al., 2019; Masiokas et al., 2020). Glacier volume loss has helped modulate river discharges, mainly in dry seasons (*e.g.*, Baraer et al., 2012; Soruco et al., 2015; Guido et al., 2016; Ayala et al., 2020).

Few studies have estimated glacier changes and their effects on hydrology using observation or modeling focused on specific Andean catchments. For instance, the global-scale study by Huss and Hock (2018) comprised 12 Andean catchments (1980-2100). They defined glacier runoff as all the melt water and rainfall coming from the initially glacierized area as given by the Randolph Glacier Inventory version 4.0. and found an increase in glacier runoff in the Tropical and Dry Andes until 2020, but a more contrasted signal in the Wet Andes: no glacier runoff change was observed in some catchments, whereas others showed a reduction or an increase. However, their estimations overlook the diversity in climate conditions and glacier morpho-topography across the Andes and inside large catchments (Caro et al., 2021): such as, latitudinal and/or longitudinal climate variations and glacier characteristics (glacier size, slope and aspect). This affects the simulation results, as they heavily rely on climate inputs and calibrated parameters. For instance, varying temperature lapse rates could result in significant disparities in glacier melt and the determination of solid/liquid precipitation on glaciers (Schuster et al., 2023). Furthermore, the selection of precipitation factor values is also crucial. Based on local studies, the glacier runoff contribution (glacier runoff relative to the total catchment runoff) in the Tropical Andes was estimated to be around 12% and 15% in the Río Santa (9°S) and La Paz (16°S) catchments, respectively (Mark and Seltzer, 2003; Soruco et al., 2015). For the La Paz catchment, Soruco et al. (2015) found no change in the glacier runoff contribution for the period 1997-2006 compared with the longer 1963-2006 period. This was attributed to the fact that the glacier surface reduction over the time-period was compensated by their increasingly negative mass balance. In the Dry Andes, the Huasco (29°S), Aconcagua (33°S) and Maipo (34°S) catchments showed a glacier runoff contribution comprised between 3 and 23% for different catchment sizes between 241 and 4843 km$^2$ (Gascoin et al., 2011; Ragettli and Pellicciotti, 2012; Ayala et al., 2020). These catchments had mainly negative glacier mass balances which were slightly interrupted during El Niño episodes (2000-2008 period), thereby reducing glacier runoff. In the Wet Andes, Dussaillant et al. (2012) estimated that some catchments in the Northern Patagonian Icefield are strongly conditioned by glacier melting. In addition, Huss and Hock (2018) did not identify changes in the glacier runoff of the Baker catchment since 1980-2000. However, these studies focused on a restricted number of catchments, employing diverse input data and methodologies over different periods. As such, these local estimations may not be indicative of the broader trends across the entire Andean region. Notably, even neighboring glacierized catchments can exhibit substantial variations in climatic and topographic characteristics (Caro et al., 2021).

Nowadays, the availability of global glaciological products such as glacier surface elevation differences and glacier volume estimation (Farinotti et al., 2019; Hugonnet et al., 2021; Millan et al., 2022) allows for large-scale glacio-hydrological simulations with the possibility to accurately calibrate and validate numerical models at the glacier scale. In addition, modeling frameworks such as the Open Global Glacier Model (OGGM, Maussion et al., 2019) have been implemented to simulate the glacier mass balance and glacier dynamics at a global scale. Therefore, OGGM and the glaciological global dataset, in combination with *in situ* meteorological and glaciological measurements, considering the differences of Andean glaciological zones, can be used to quantify the glacier retreat and the related hydrological responses at the catchment scale across the Andes, while taking

the related uncertainties into account. Currently, reconstructions of glacier surface mass balance across the Andes
(9-52°S) rely on a temperature-index model. Notably, higher mean melt factor values are identified in the
Tropical Andes (0.3-0.5 mm h$^{-1}$ °C$^{-1}$), compared to the Dry Andes (0.3-0.4 mm h$^{-1}$ °C$^{-1}$) and Wet Andes (0.1-0.5
mm h$^{-1}$ °C$^{-1}$) (*e.g.*, Fukami & Naruse, 1987; Koisumi and Naruse, 1992; Stuefer et al., 1999, 2007; Takeuchi et
al., 1995; Rivera, 2004; Sicart et al., 2008; Condom et al., 2011; Caro, 2014; Huss and Hock, 2015; Bravo et al.,
83 2017).

Here, using OGGM, we estimate the glacier changes (area and volume) and the consecutive hydrological
responses called glacier runoff (which is composed of glacier melt [ice melt and snow melt] and rainfall on
glaciers) for 786 catchments across the Andes (11°N-55°S) with a glacierized surface of at least 0.01% for the
period 2000-2019. This approach allows us to study the behavior of glaciers across the entire Andes and within
specific catchments (for instance those previously studied). Considering the significant hydro-glaciological
variations in neighboring catchments and the potential biases within climatic datasets, the air temperature and
precipitation data from the TerraClimate dataset (Abatzoglou et al., 2018) were corrected using *in situ* data
across the Andes. On the other hand, the simulation procedure considered the calibration of glacier surface mass
balance and glacier volume. Both, corrections of climate as well as calibrations (at the glacier scale) were
performed considering the climatic and morphometric differences in the Andes, represented through the
glaciological zones (Caro et al., 2021). Given that the most important uncertainties in simulating future glacier
evolution come, among other factors, from the implementation of the models during the historical period, we
validate our simulation and calibration outcomes against observed data from glaciers and catchments.
Section 2 presents the data and methods. In Section 3, we describe the glacier changes and hydrological
responses at the glaciological zone and catchment scales across the Andes. In Section 4, we discuss our results
and the main steps forward compared to previous research.

## 2 Data and methods

This section comprises the processed data used as input and during the modeling framework (Figure 1).

## 2.1 Data collection and preprocessing

### 2.1.1 Historical climate data

We used two climate datasets: the TerraClimate reanalysis (Climate box in Figure 1) and *in situ* measurements
from meteorological stations (*in situ* measurements box in Figure 1). TerraClimate is based on reanalysis data
since 1958, with a 4 km grid size at a monthly time scale, and was validated with the Global Historical
Climatology Network (temperature, r = 0.95, MAE = 0.32°C; precipitation, r = 0.9, MEA = 9.1%) (Abatzoglou
et al., 2018). The mean temperature was estimated from the maximum and minimum temperature whereas
precipitation data is accumulated on a monthly basis. The meteorological records were compiled from Andean
organizations and scientific reports (Rabatel et al., 2011; MacDonnell et al., 2013; Schaefer et al., 2017; CECs,
2018; Shaw et al., 2020; Hernández et al., 2021; CEAZA, 2022; DGA, 2022; GLACIOCLIM, 2022; IANIGLA,
2022; Mateo et al., 2022; Senamhi, 2022). The mean monthly air temperature measurements were taken from 35

off-glacier and on-glacier meteorological stations, the latter being rare, located between 9 and 51°S. However, it is important to note that long-term measurements were not available northward of 9°S (the Inner Tropics, IT). To address this, data from stations located in the Outer Tropics (OT) were used as a reference for temperature corrections in this zone, which could affect the performance in the estimation of calibrated parameters such as the melt factor. The location and main properties of the meteorological stations are shown in Supplementary Table S1.

**2.1.2 Climatic data correction and evaluation**

For the temperature variable, we first quantified the local vertical annual temperature lapse rates using the *in situ* measurements for 33 sites across the Andes (see Table and Figure S1). Then, the TerraClimate temperature was corrected with these *in situ* records so that they could be used in the simulations (correction box in Figure 1). Last, the corrected TerraClimate temperature was evaluated via a comparison with the 34 situ data (evaluation box in Figure 1). Conversely, the precipitation variable from the TerraClimate reanalysis was scaled using the mass balance measurements for 10 monitored glaciers and was evaluated for 15 glaciers (correction box in Figure 1). Specific data is available in Tables S3, S4 and S5.

Vertical temperature lapse rates (temperature LRs) from the *in situ* records were estimated for each glaciological zone across the Andes as per Gao et al. (2012). The temperature LRs are presented in Figure S1. These gradients were applied to correct the raw TerraClimate temperature on the glaciers ($rTC_t$). The corrected TerraClimate temperature at the mean elevation of glacier ($cTC_t$) was calculated using the following equation:

$$cTC_t = rTC_t + \Gamma * \Delta h , \tag{1}$$

where $\Gamma$ is the temperature LR estimated here, and $\Delta h$ is the elevation difference between a glacier elevation and the mean elevation of the TerraClimate grid-cell where the glacier is located.

Then, we assessed the $cTC_t$ in meteorological station locations (9°S-51°S) on a monthly scale, paying attention to the monthly variability in temperature as well as to the mean temperature for all the periods with data. The $cTC_t$ monthly mean variability was evaluated using the Pearson correlation coefficient, whereas the mean temperature for the whole period considered the mean difference between $cTC_t$ and the observed temperature (biases).

In addition, the total precipitation was scaled ($cTC_p$) using precipitation factors ($Pf$) for each glaciological zone across the Andes (see the relationship between solid precipitation and Pf in equation 3). In a second step we discriminate snowfall and rainfall using a linear regression between the temperature thresholds to obtain the solid/liquid precipitation fraction (Maussion et al., 2019). We ran 31 simulations for 18 glaciers with mass balance measurements across the Andes using *Pf* values between 1 and 4 taking previous studies into account (Masiokas et al., 2016; Burger et al., 2019; Farías-Barahona et al., 2020). Ultimately, 10 glaciers were selected (see Table S3), because their simulated mass balances showed a closer standard deviation in comparison with measurements. The goal was to find the closest simulated mass balance standard deviation ($simSD_{mb}$) in comparison with the measured mass balance standard deviation ($obsSD_{mb}$) using different *Pf* values (Equation 2).

$$Pf = \{ 1 \leq Pf \leq 4 \; : \; simSD_{mb} \approx obsSD_{mb} \} , \qquad (2)$$

A similar methodology was proposed by Marzeion et al. (2012) and Maussion et al. (2019). The results of the closest simulated mass balance standard deviations and associated *Pf* are presented in Supplementary Table S3. The simulated annual mass balance was evaluated on 15 monitored glaciers using a Pearson correlation coefficient and bias (as the average difference) from simulated mass balance and measured mass balance (evaluation box in Figure 1). In addition, details such as snow/rainfall partitioning are described hereafter and in the model implementation (Section 2.2).

## 2.1.3 Glacier data

**Glacier inventory**

We used version 6.0 of the Randolph Glacier Inventory (RGI Consortium, 2017) to extract the characteristics of each glacier, *e.g.*, location, area, glacier front in land or water (glacier inventory box in Figure 1). The RGI v6.0 was checked using the national glacier inventories compiled by Caro et al. (2021), filtering every RGI glacier that was not found in the NGI, to obtain a total glacierized surface area of 30,943 km$^2$ (filtering 633 km$^2$). The glacier extent in the RGI v6.0 is representative of the early 2000s. The analysis by catchment and glaciological zones is related to the locations and elevation of these glaciers. Overall, 36% of the total glacierized surface area across the Andes is considered. Over 85% of the glacierized surface area in the Dry Andes (18°S-37°S) and 79% in the Tropical Andes (11°N-18°S) are considered, which corresponds to 11% (3,377 km$^2$, in 321 catchments) of the total glacierized area of the Andes. For the Wet Andes (37°S-55°S), 29% of the glacierized surface area in the region is considered, which corresponds to 26% (7,905 km$^2$, in 465 catchments) of the total glacierized area in the Andes (see the distribution of the catchments in Figure 2a). The simulated glacierized surface area is lower in the Wet Andes due to the filtering out of the numerous calving glaciers found there.

**Glacier mass balance**

The mass balance datasets were comprised of the global glacier surface elevation change product of Hugonnet et al. (2021) (calibration box in Figure 1) and *in situ* measurements of the glacier surface mass balance (evaluation box in Figure 1) since 2000 from different institutions (*e.g.*, Marangunic et al., 2021; WGMS, 2021). Hugonnet et al. (2021) product was quantified for each glacier using the OGGM toolbox (Figure 2d). Then, the geodetic mass balance estimates were obtained for every glacier of the RGI v6.0. *In situ* measurements of the glacier surface mass balance are available between 5°N and 55°S (across all Andean regions) at the hydrological year scale (dates vary according to the latitude). However, the Tropical Andes is represented by just two glaciers (Conejeras and Zongo glaciers), producing an underrepresentation in the evaluation of the simulated mass balance in this region. The location and main characteristics of the 18 monitored glaciers are shown in Supplementary Table S4.

**Glacier volume**

The global glacier ice thickness product of Farinotti et al. (2019) was used to calibrate each glacier of the RGI v6.0 in OGGM (calibration box in Figure 1). Farinotti et al. (2019) pooled the outputs of five different models to determine the distribution of the ice thickness on 215,000 glaciers outside the Greenland and Antarctic ice sheets.

**2.1.4 Glaciological zones and catchments**

Eleven glaciological zones across the Andes were compiled from Caro et al. (2021) and all glaciers northward of the Outer Tropics were considered as zone number 12, called the Inner Tropics. To identify the glacierized area in each catchment, a spatial intersection was made between the glaciers identified in the filtered RGI v6.0 and the Level 9 HydroSHEDS catchments (Lehner et al., 2006). Then, we considered catchments with a glacierized surface area >= 0.01% (max = 62%, mean = 5%, median = 2%). We selected 786 catchments with a surface area between 3,236 and 20 km$^2$ across the Andes (11°N-55°S), including 13,179 glaciers with a total surface area of 11,282 km$^2$ (36% of the total glacierized surface area in the Andes).

Calving glaciers (lake- and marine-terminating, 15,444 km$^2$), primarily located in the Northern and Southern Patagonian Icefields and in the Cordillera Darwin, were not considered because the calving process implemented in this version of OGGM (1.5.3) which relies on Hugonnet et al. (2019) data to calibrate the simulated mass balance, could exhibit significant uncertainty when applied to these particular glaciers. In this regard, Zhang et al. (2023) estimated an underestimation of glacier mass loss for lake-terminating glaciers using geodetic methods, accounting for a subaqueous mass loss of $10 \pm 4\%$ in the central Himalaya during the period 2000 to 2020. Their findings revealed that the total mass loss for certain glaciers was underestimated by as much as $65 \pm 43\%$. The glaciers that were not simulated for the internal model inconsistencies account for less than 1% of the total glacierized surface area. The other remaining 4,514 km$^2$ filtered glacierized surface area corresponds to glacierized catchments that present an increase in glacier volume but a reduction in the glacierized surface area. Only 59 km$^2$ was associated with glaciers filtered in the Outer Tropics 1 zone.

We selected the La Paz (Soruco et al., 2015), Maipo (Ayala et al., 2020) and Baker (Dussaillant et al., 2012) catchments located in glaciological regions with different climatic and morphometric characteristics (Caro et al., 2021) to evaluate our simulations in terms of glacier changes and glacier runoff contributions over the period 2000-2019. In the La Paz and Maipo catchments, previous hydro-glaciological studies have quantified the impact of glacier changes and their hydrological contribution. However, these studies often overlook relevant processes such as variations in precipitation, temperature corrections, and the simulation of glacier dynamics. On the other hand, in the Baker catchment, there are currently no estimations of glacier runoff contributions. These three catchments allow us to make comparisons with our regional simulations at the Andes scale using consistent data (*e.g.*, corrected climate datasets and glacier outlines) and methods (*e.g.*, simulating mass balance, dynamics, and glacier runoff), update previous results, and provide new glacier runoff estimates. For example, it is necessary to understand what occurs during the prolonged dry period in Central Chile and Argentina. In addition, river discharge records were collected from Soruco et al. (2015) and the CAMELS-CL project (Alvarez-Garreton et al., 2018) for Bolivia and Chile, respectively. In Bolivia, we considered the four glacierized head catchments providing water to the La Paz catchment: Tuni-Condoriri, Milluni, Hampaturi and Incachaca (discharge records from 2001 to 2007) with a total surface area of 227 km$^2$ and 7.5% of the glacierized surface area (mean elevation

of 5,019 m a.s.l.). In Chile, for the Maipo catchment, we compiled records from Río Maipo at the El Manzano station (catchment id = 5710001; 4839 km$^2$; discharge records from 1990 to 2019) and Río Mapocho at the Los Almendros catchments (catchment id = 5722002; 638 km$^2$; discharge records from 1990 to 2019) with a glacierized surface area of 7.5% (mean elevation of 4,259 m a.s.l.). For the Baker catchment, we used the Río Baker Bajo Ñadis records (catchment id = 11545000; 27403 km$^2$; discharge records from 2004 to 2019), considering a glacierized surface area of 8.2% (mean elevation of 1,612 m a.s.l.). Note that only the glacier runoff contribution will be simulated.

## 2.2 OGGM details

We ran the OGGM model (Maussion et al., 2019) for each glacier and then the results per catchment were aggregated for each of the 786 catchments across the Andes (including the three selected test catchments for a detailed analysis). OGGM is a modular and open-source numerical workflow implemented in Python that provides pre-processed datasets such as DEMs, glacier hypsometry, glacier flowlines, etc. that can be used to explicitly simulate glacier mass balance and ice dynamics using calibrated parameter values for each glacier. Here, we ran OGGM from Level 2, comprising the flowlines and their downstream lines. However, we used a new baseline climate time series (corrected TerraClimate) as input data. We also calibrated the mass balances and the bed inversion (ice thickness) that allowed us to obtain hydrological outputs (glacier runoff) (details in https://docs.oggm.org/en/v1.4.0/input-data.html). The spatio-temporal configuration of the model used in this study is at the glacier scale and at the monthly time step. The simulation results were analyzed at different spatial scales: by glacierized catchment, glaciological zone, and regionally.

The required input data for running the model are as follows: air temperature and precipitation time series, and glacier outlines and surface topography for specific years. From these input data we computed annual simulated processes such as the surface mass balance, glacier volume and area, monthly glacier melt (snow and ice) and rainfall on glaciers (Figure 1). Modeled processes such as the surface mass balance and glacier volume were calibrated (Table 1 and Figure 2). The calibration procedure of the parameters was applied per glacier to match the simulated mass balance 2000-2019 to the geodetic mass balance product from Hugonnet et al. (2021). The simulated glacier volume was calibrated using Farinotti et al. (2019) product at a glaciological zone scale to fit the Glen A parameter. In other words, the same Glen A parameter was used for each glaciological zone.

First, using a glacier outline and topography, OGGM estimates the flow lines and catchments per glacier, and then the flow lines are calculated using a geometrical algorithm (adapted from Kienholz et al., 2014). Assuming a bed shape, it estimates the ice thickness based on mass conservation and shallow-ice approximation (Farinotti et al., 2009; Maussion et al., 2019). After these numerical steps, it is possible to determine the area and volume per glacier. The mass balance is implemented using a precipitation phase partitioning and a temperature-index approach (Braun and Renner, 1992; Hock, 2003; Marzeion et al., 2012). The monthly mass balance $mb_i$ at an elevation z is calculated as follows:

$$mb_i(z) = TC_{p\,i}^{snow}(z) * P_f - M_f * max\left(cTC_{t\,i}(z) - T_{melt}, 0\right), \qquad (3)$$

where $TC_{p\,i}^{snow}$ is the TerraClimate solid precipitation before being scaled by the precipitation correction factor (
$P_f$), $M_f$ is the glacier's temperature melt factor, $cTC_{t\,i}$ is the monthly corrected TerraClimate temperature. $P_f$ and
$M_f$ parameters are related to the snow/ice onset ($T_{melt}$) and precipitation fraction ($T_i^{snow}$ and $T_i^{rain}$). Their values
are different across the Andes. $T_{melt}$ is the monthly air temperature above which snow/ice melt is assumed to
occur (0°C for the Dry and Wet Andes and 2.1°C for the Tropical Andes). $T_i^{snow}$ is calculated as a fraction of the
total precipitation ($cTC_p$) where 100% is obtained if $cTC_{t\,i} <= T_i^{snow}$ (0°C for the Dry and Wet Andes and 2.1°C
for the Tropical Andes) and 0% if $cTC_{t\,i} >= T_i^{rain}$ (2°C for the Dry and Wet Andes and 4.1°C for the Tropical
Andes), using a linear regression between these temperature thresholds to obtain the solid/liquid precipitation
fraction. Here, $M_f$ was calibrated for each glacier individually using the previously described glacier volume
change datasets (Hugonnet et al., 2021). The calibrated parameter values are summarized by glaciological zone
in Table 1.

| Table 1. Calibrated parameter values used in the glacier mass balance and volume simulations across the Andes (11°N-55°S) during the period 2000-2019 | | | | | | | |
|---|---|---|---|---|---|---|---|
| | | Mass balance parameter values | | | | | Volume parameter |
| Region | Zone | Temperature LR [°C/m] | Precipitation factor [-] | Mean melt factor [mm mth$^{-1}$ °C$^{-1}$] | Temperature for melt onset [°C] | Temperature at start of snowfall [°C] | Temperature at start of rainfall [°C] | Glen A inversion [s$^{-1}$ Pa$^{-3}$] |
| Tropical Andes | IT* | -0.0066 | 1 | 434 | 2.1 | 2.1 | 4.1 | 2.4 10$^{-23}$ |
| | OT2* | | | 284 | | | | 6.3 10$^{-24}$ |
| | OT3* | | | 432 | | | | 1.2 10$^{-23}$ |
| Dry Andes | DA1 | -0.0082 | 2.8 | 418 | 0 | 0 | 2 | 2.4 10$^{-25}$ |
| | DA2 | -0.0065 | 1.9 | 479 | | | | 1.3 10$^{-23}$ |
| | DA3 | -0.0063 | 4 | 299 | | | | 2 10$^{-24}$ |
| Wet Andes | WA1 | -0.0051 | 4 | 103 | | | | 1.7 10$^{-23}$ |
| | WA2 | | 4 | 118 | | | | 1.9 10$^{-23}$ |
| | WA3 | -0.0063 | 2.3 | 152 | | | | 6 10$^{-24}$ |
| | WA4 | | | 128 | | | | 1.3 10$^{-23}$ |
| | WA5 | | | 179 | | | | 1 10$^{-23}$ |
| | WA6 | | | 139 | | | | 1.5 10$^{-23}$ |
| *related to zones of the Tropical Andes region: Inner Tropics (IT) and Outer Tropics (OT2 and OT3) | | | | | | | |


## 2.2.1 Model setup, calibration and validation

The input data are as follows: the corrected monthly TerraClimate precipitation ($cTC_p$) and temperature ($cTC_t$),
glacier outlines were obtained from RGI v6.0 (RGI Consortium, 2017), assuming the glacier outlines of all
glaciers were made for the year 2000. The surface topography data were sourced from NASADEM (Crippen et
al., 2016). NASADEM has a spatial resolution of 1 arcsecond (~30m), and the data were acquired in February
2000 (NASA JPL, 2020). The simulated glacier volume was calibrated using Farinotti et al. (2019) product at a
glaciological zone scale to fit the Glen A parameter (Table 1 and Figure 2).
Last, the simulated mass balance was evaluated in comparison with *in situ* mass balance observations
(Marangunic et al., 2021; WGMS, 2021). Although the OGGM outputs are in calendar years and the
observations are in hydrological years, we consider it essential to evaluate the interannual performance (Pearson
correlation, p-value, variance, RMSE and bias from average difference) and the cumulative mass balance since
the year 2000.

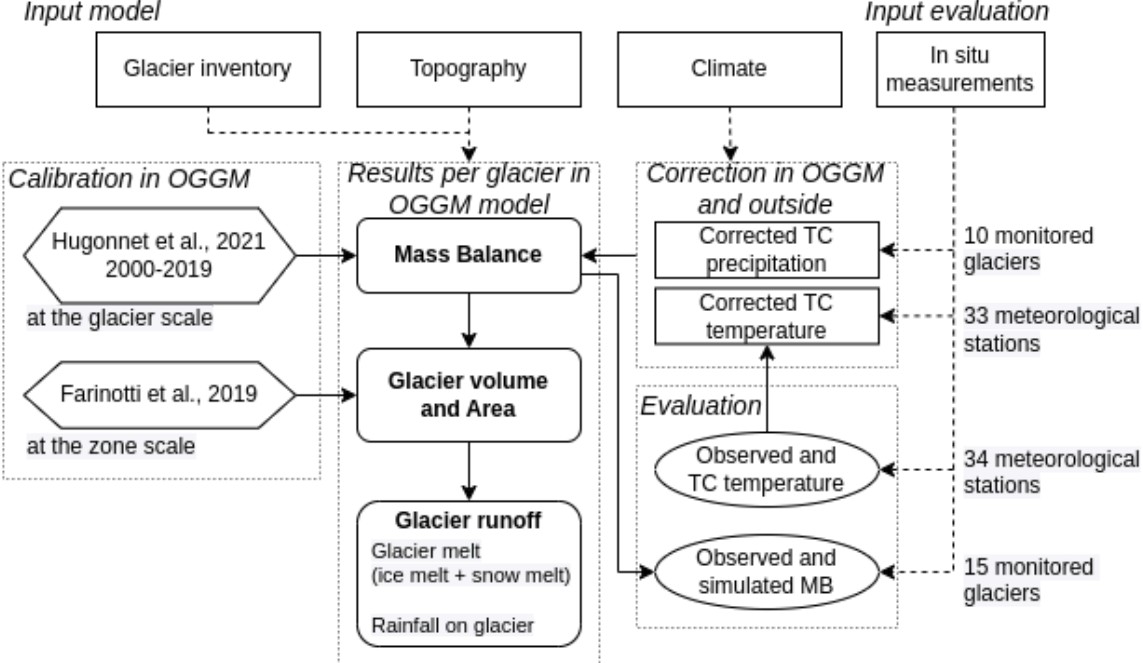

**Figure 1. Workflow per simulated glacier using OGGM between 2000 and 2019. Two groups of input data were used: one to run OGGM and the second to correct/evaluate the TerraClimate temperature (cTCt) and precipitation (cTCp). Then, the mass balance and glacier volume were calibrated. Lastly, results such as the cTCt and glacier mass balance were evaluated at 34 meteorological stations and on 15 glaciers with mass balance observations. The corrections in OGGM and outside box refer to analyses performed by running the model and also analyzing data outside the model tool. An example is the estimation of temperature lapse rates, which were estimated from *in situ* measurements but introduced in the OGGM model as a parameter value.**

**3 Results**
**3.1 Climatic variations on glaciers across the Andes during the period 2000-2019**
The climate associated with 786 Andean glacierized catchments (11°N-55°S) presents a mean corrected
TerraClimate temperature (cTCt) of -0.2 ± 2.2°C and a mean annual corrected TerraClimate precipitation (cTCp)
of 2699 ± 2006 mm yr$^{-1}$ between 2000 and 2019. The various glaciological regions show significant climatic
differences, with contrasting extreme values between the Tropical Andes and Wet Andes in terms of mean
annual precipitation (939 ± 261 mm yr$^{-1}$ and 3751 ± 1860 mm yr$^{-1}$, respectively) and mean annual temperature
between the Dry Andes and Tropical Andes (-3.7 ± 1.4°C and 1.3 ± 0.8°C, respectively). Certain glaciological

zones highlight very negative and positive mean annual temperature values such as Dry Andes 2 (-4.8°C) and Wet Andes 2 (1.9°C) and lower and higher cumulative precipitation values such as Dry Andes 1 (447 mm yr$^{-1}$) and Wet Andes 5 (6075 mm yr$^{-1}$). Meanwhile, variations in climate between the periods 2000-2009 and 2010-2019 across the Andes show a cumulative precipitation decrease in -9% (-234 mm yr$^{-1}$) and a mean annual temperature increase in 0.4 ± 0.1°C. Between these two periods, precipitation is decreasing primarily in the Dry Andes (-256 mm yr$^{-1}$; -23%) and Wet Andes (-337 mm yr$^{-1}$; -9%), and increasing in the Tropical Andes (44 mm yr$^{-1}$; 5%), whereas the temperature is increasing between 0.3-0.4°C in all regions. At the glaciological zone scale, only the Tropical Andes and Dry Andes 1 (12%) show a cumulative increase in precipitation, whereas a larger decrease in precipitation is found in  Dry Andes 2 (-32%) and Dry Andes 3 (-27%). The mean annual temperature increases in all zones, especially the Inner Tropics (+0.6°C) followed by Wet Andes 3 (+0.5°C). A summary of variations in climate by glaciological zone is presented in Table 2.

Our cTCt evaluation is statistically significant (p-value < 0.01) at 32 meteorological stations with a mean temperature bias of 0.4°C and a mean correlation of 0.96 (These results are available in Table S2 and Figure S2). The regional results show a larger bias in the Tropical Andes (mean = 2.1°C, four stations) with a meteorological station mean elevation of 4,985 m a.s.l., where cTCt cannot represent the mean monthly temperature. However, cTCt well represents the maximum temperatures in spring/summer and the minimum temperatures in winter. The lowest bias is observed in the Wet Andes and Dry Andes. The Wet Andes, with a meteorological station mean elevation of 813 m a.s.l., shows good results in terms of reproducing the mean monthly temperature in most stations, with a minimum correlation higher than 0.86. In the Dry Andes, with a meteorological station mean elevation of 3,753 m a.s.l. (18 stations) and bias of 0.2°C, the cTCt reproduces the mean monthly temperature very well. However, in some stations such as La Frontera and Estrecho Glacier (29°S), the mean cTCt is warmer than 6°C, whereas in other stations such as El Yeso Embalse (33.7°S) and Cipreses glacier (34.5°S), the mean cTCt is colder than 6°C. The detailed cTCt evaluation based on bias and Pearson's correlation can be found in Tables S1 and S3 and Figure S2 of the Supplementary Materials. The cTCt presented a mean bias of 2.1°C in the Tropical Andes and a mean bias of 0.2°C in the Dry Andes and Wet Andes in comparison with *in situ* measurements.

**3.2 Glaciological changes across the Andes during the period 2000-2019**

The 36% of the total glacierized surface area across the Andes (11°N-55°S) are simulated to obtain annual glacier area and glacier volume, as well as the monthly glacier runoff (glacier melting and rainfall on glaciers).

Considering mean values for the periods 2000-2009 and 2010-2019, the glacier volume and surface area in the Andean catchments show a decrease by -8.3% (-59.1 km$^3$) and -2.2% (-245 km$^2$), respectively. This corresponds to a mean annual mass balance difference between the two periods of -0.5 ± 0.3 m w.e. yr$^{-1}$ (Figure 2d). A decrease in glacier volume and surface is seen in 93% of the catchments (n = 724) whereas 7% of the catchments (n = 65) present an increase in glacier volume and surface. The loss in glacier volume (Figure 2b) is largest (-47.8 km$^3$, -9%) in the Wet Andes, followed by the Tropical Andes (-5.9 km$^3$, -7%) and Dry Andes (-5.4 km$^3$, -6%). Similarly, a larger decrease in the glacier surface area (Figure 2c) is observed in the Wet Andes (-144.4 km$^2$, -2%), followed by the Tropical Andes (-55.5 km$^2$, -4%) and lastly the Dry Andes (-45.2 km$^2$, -3%). As expected, the correlation between both glacier change variables is consistent at the zone scale, showing a positive correlation between the changes in area and volume (r = 0.9).

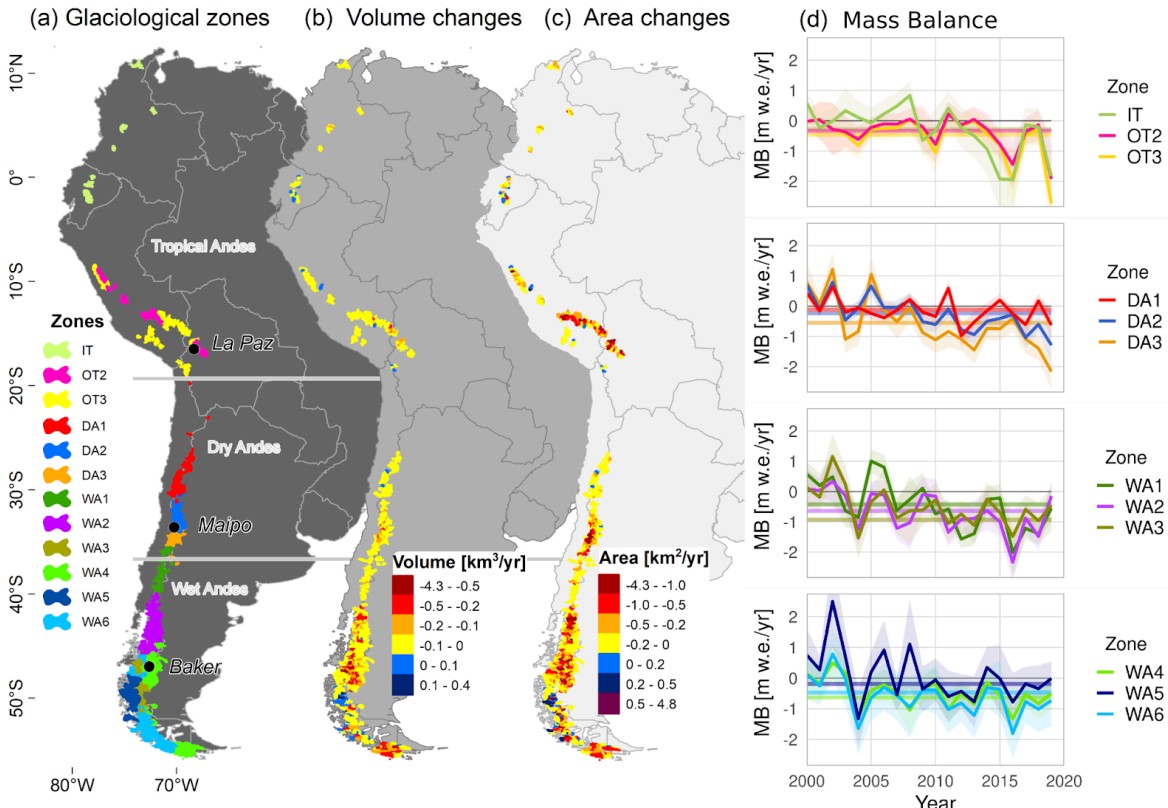

**Figure 2. Recent glacier changes across the Andes. The glacier changes represent the mean annual differences between the periods 2000-2009 and 2010-2019 per catchment (n = 786). (a) It shows the distribution of the glaciological zones (11°N-55°S), followed by the (b) volume and (c) area changes at the catchment scale. The (d) annual specific mass balances are presented in each glaciological zone (the shaded areas are the standard deviation), where the straight lines correspond to the mean geodetic mass balance (2000-2019) estimated by Hugonnet et al. (2021).**


When estimating the mass balance, it is interesting to check the calibrated melt factors ($M_f$) of the temperature
index-model in order to evaluate its possible regionalization, i.e. to evaluate the spatial coherence (see Table 1
and Figure 3). The mean calibrated melt factor values decrease from the Tropical Andes toward the Wet Andes
(Tropical Andes = 0.5 ± 0.3 mm h$^{-1}$ °C$^{-1}$, Dry Andes = 0.6 ± 0.2 mm h$^{-1}$ °C$^{-1}$, Wet Andes = 0.2 ± 0.1 mm h$^{-1}$ °C$^{-1}$).
The lowest mean temperatures estimated in the Dry Andes imply higher factor values to reach the calibrated
mass loss in the few months in which the temperatures exceed 0°C. The opposite can be observed in the Wet
Andes, where low factor values are associated with a greater number of months with temperatures exceeding
0°C. We obtain very similar values in contiguous zones, with the lowest values found in the Wet Andes (mean
below 179 mm mth$^{-1}$ °C$^{-1}$), followed by the Tropical Andes (mean below 434 mm mth$^{-1}$ °C$^{-1}$), and the Dry Andes
(mean below 479 mm mth$^{-1}$ °C$^{-1}$). The largest melt factor values are found in the Dry Andes where the Dry
Andes 2 zone (mean = 479 mm mth$^{-1}$ °C$^{-1}$) presents the lowest mean temperatures across the Andes (-4.8°C
between 2000-2019). The lowest melting factor values are calibrated in the Wet Andes where zone Wet Andes 1
(mean = 103 mm mth$^{-1}$ °C$^{-1}$) shows high mean temperatures (1.8°C between 2000-2019). Despite this, a lower
correlation between the melt factors and mean temperature for the 2000-2019 period is estimated (r = -0.5;
p-value = 0.08). Conversely, the correlation between the melt factors and mean precipitation for the 2000-2019
period is high (r = -0.8; p-value = 0.002).
To test our results we evaluated the simulated mass balance evaluation for the 15 glaciers that can be found in
Tables S4, S5 and Figures S3 and S4 of the Supplementary Materials. The *in situ* data show a mean negative
mass balance (-832 ± 795 mm w.e. yr$^{-1}$) between 2000 and 2019 greater than our mean simulated mass balance
(-647 ± 713 mm w.e. yr$^{-1}$) in the same glaciers. The evaluation results give a mean Pearson correlation of 0.67
(except for Agua Negra, Ortigas 1, Guanaco and Amarillo glaciers, which shows either no correlation or a
negative correlation) with an underestimation of the mean simulated mass balance of 185 mm w.e. yr$^{-1}$ (bias);
40% of the glaciers present a correlation equal to or greater than 0.7. In terms of the best results by glaciological
region, in the Tropical Andes, the Conejeras glacier has a high correlation (r = 0.9) and bias (1104 mm w.e. yr$^{-1}$),
whereas in the Dry Andes, the Piloto Este, Paula, Paloma Este and Del Rincón glaciers display a high correlation
(r >= 0.8) and a mean bias of 351 mm w.e. yr$^{-1}$. In the Wet Andes, the Mocho-Choshuenco and Martial Este
glaciers show a moderate correlation (r = 0.5) and a lower overestimation of the simulated mass balance (-118
mm w.e. yr$^{-1}$). Model limitations are observed on the Zongo glacier (r = 0.3 and bias = -224 mm w.e. yr$^{-1}$) in the
Tropical Andes. In the Dry Andes 1, no correlation is observed in the three monitored glaciers (Guanaco,
Amarillo and Ortigas 1); this is mainly because sublimation is very high on these glaciers, reaching 81% of the
annual ablation (MacDonell et al., 2013). On the other hand, sublimation is lower southward in the Dry Andes 2
with 7% of the annual ablation (Ayala et al., 2017). For the tropical zone, sublimation is close to 13% in Outer
Tropics (Sicart et al., 2005) and 5% in Inner Tropics (Favier et al., 2004). However, sublimation is implicitly
included in the model through the calibrated melt factor values, which are derived from measured mass balance
data by Hugonnet et al. (2021). As a result, our estimates of snow/ice melt in the Dry Andes 1 zone tend to be
overestimated.
The details of the glacier changes in the 786 Andean catchments, which are larger in the Wet Andes followed by
the Tropical Andes and then the Dry Andes, are available in the Supplementary Materials.

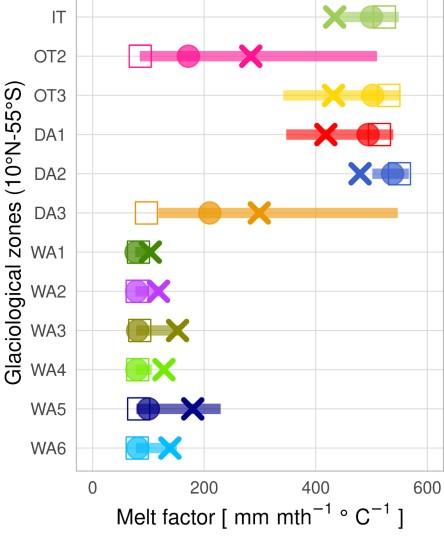

**Figure 3. Statistics for the calibrated melt factors per glacier at the glaciological zone scale across the Andes. This figure shows the mean (x), median (circle), mode (square), and percentile 25 and 75 (lines) values for 13,179 glaciers.**


### 3.3 Changes in glacier runoff across the Andes during the period 2000-2019

Due to glacier changes across the Andes, high glacier runoff variations are observed from glacier melt and rainfall on glaciers (Figure 4). The mean annual glacier melt in all catchments for the period 2000-2019 was 696 $m^3/s$. At the regional scale, the Wet Andes shows the largest mean annual glacier melt in the Andes (583.5 $m^3/s$), followed by the Dry Andes (59.9 $m^3/s$) and then the Tropical Andes (52.7 $m^3/s$). However, if we look at the mean annual glacier melt changes between the periods 2000-2009 and 2010-2019, we see an increase of 12% (86.5 $m^3/s$) across the Andes, where 84% (n = 661) of catchments show an increase and 12% (n = 95) of them present a decrease. As Table S6 shows, an increase in glacier melt is observed in catchments with a higher glacier elevation, larger glacier size, lower mean temperature and higher mean precipitation compared with catchments that show either a decrease in glacier melt or no changes at all. These latter catchments also show the largest decrease in precipitation (-10 to -14%).

The mean annual glacier melt changes show the largest percentage increase in the Tropical Andes (40%, 21 $m^3/s$), followed by the Dry Andes (36%, 21.7 $m^3/s$), and the Wet Andes (8%, 4.8 $m^3/s$). In addition, significant differences are observed for the different zones: for instance, the Inner Tropics in the Tropical Andes presents the largest increase (73% with only 4.1 $m^3/s$), followed by Dry Andes 1 (62% with only 1.8 $m^3/s$) in the Dry Andes. In the Wet Andes, the larger percentage of increase in the mean annual glacier melt changes is observed in Wet Andes 5 (14% with 4.1 $m^3/s$), showing a lower percentage in comparison with the Inner Tropics and Dry Andes 1 zones, however, its absolute increase in glacier melt is equal to or greater than 4.1 $m^3/s$. These results per glaciological zone are summarized in Table 2. Related to the previously described glacier changes (see Section 3.2) between the periods 2000-2009 and 2010-2019, at the glaciological zone scale, we logically find a high negative correlation between the glacier melt and glacier volume changes in the Tropical Andes and Dry Andes (r = -0.9) and the Wet Andes (r = -1).

In addition, the mean annual rainfall on glaciers across the Andes is 387 $m^3/s$ for the period 2000-2019. The Wet Andes has the largest amount of annual rainfall (372.7 $m^3/s$), followed by the Tropical Andes (10.5 $m^3/s$) and Dry Andes (4.2 $m^3/s$) with the lowest contribution of rainfall.

In terms of the variations in the mean annual rainfall on glaciers between the periods 2000-2009 and 2010-2019, we observe a reduction of -2% (-7.6 $m^3/s$) across the Andes, showing a reduction in 41% of the catchments (n = 322) whereas the largest proportion of the catchments (51%, n = 403) show an increase. Table S6 shows that the catchments with the larger increase of rainfall on glaciers are concentrated in the same latitude range as the catchments with an increase in glacier melt. These catchments have similar glacier elevations and glacier sizes. The catchments that do not show variations in rainfall on glaciers are concentrated in the Dry Andes region, where the rainfall contributes less to the glacier runoff volume.

At the glaciological region scale, the mean annual rainfall on glaciers decreases in the Wet Andes (-3%, 10.1 $m^3/s$), but increases in the Tropical Andes (23%, 2.4 $m^3/s$) and Dry Andes (3%, 0.1 $m^3/s$). In addition, large differences are observed in the glaciological zones (Table 2): e.g. Dry Andes 1 in the Dry Andes has the largest

 percentage increase (106% with only 0.2 m³/s), followed by Inner Tropics (74% with only 0.4 m³/s) in the

 Tropical Andes. In the Wet Andes, the larger increase (in percent) in the mean annual rain on the glaciers is

 observed in Wet Andes 5 (6.6% with 2.1 m³/s). Other zones such as Wet Andes 2 and Wet Andes 6 show large

 absolute reductions (-11.7 m³/s and -4.5 m³/s, respectively).

 The changes in glacier melt and rainfall on glaciers observed in the Tropical Andes, Dry Andes and Wet Andes

 are summarized in Table 2, and are available for the 786 Andean catchments in the Supplementary Materials.

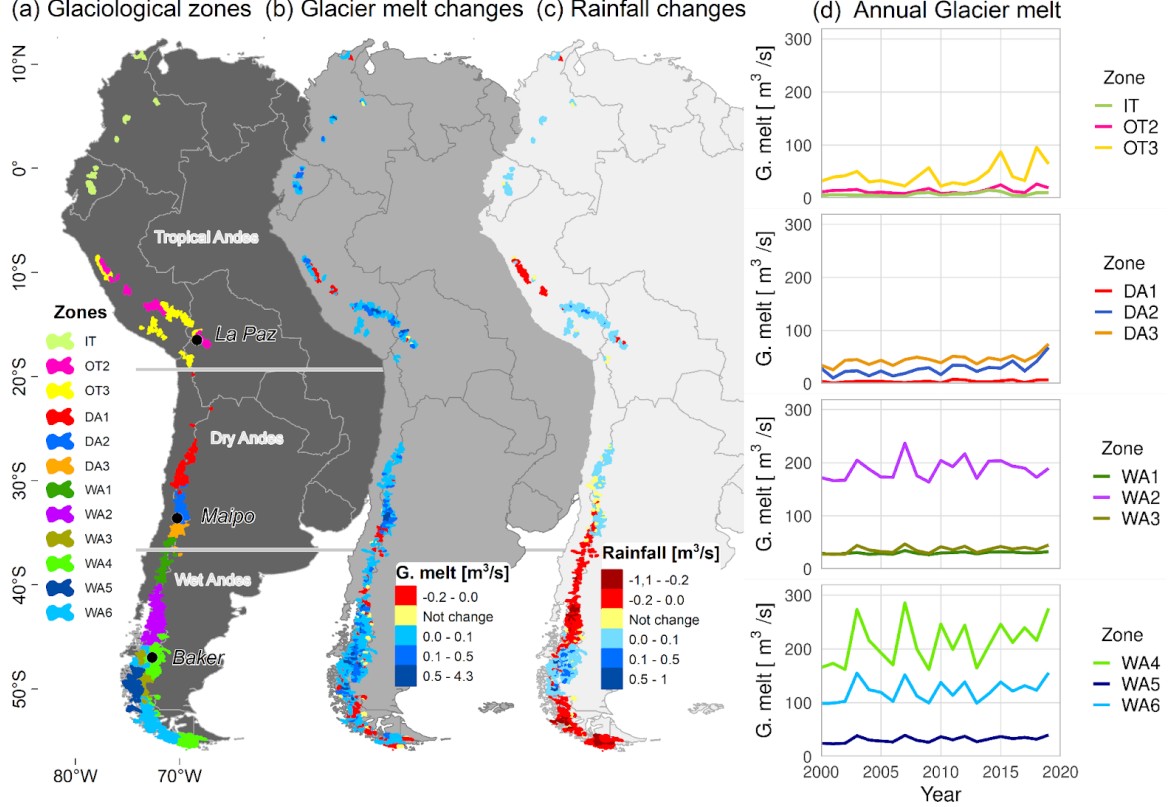

**Figure 4. Recent glacier runoff components across the Andes. The total glacier melt and rainfall on glaciers represent the mean differences between the periods 2010-2019 and 2000-2009 per catchment (n = 786). (a) It shows the distribution of the glaciological zones (11°N-55°S), followed by (b) glacier melt and (c) rainfall on glaciers at the catchment scale. The (d) total annual glacier melt is presented in each glaciological zone. G. melt and Rainfall refer to changes in (b) Glacier melt and (c) Rainfall on glaciers, respectively, meanwhile, G. melt in the Y-axis in (d) refers to cumulative annual glacier melt by glaciological zone.**

| Table 2. Mean annual changes in glacier area and volume, glacier runoff and climate between periods 2000-2009 and 2010-2019 at the glaciological zones scale across the Andes (11°N-55°S) | | | | | | | | |
|---|---|---|---|---|---|---|---|---|
| Region | Zone | Change in surface area [km²] (%) | Change in volume [km³] (%) | Change in glacier melt [m³/s] (%) | Change in rainfall on glaciers [m³/s] (%) | Simulated area [km²] and percentage in total glacierized area (%) | cTCt change [°C] | cTCp change [mm yr⁻¹ (%)] |
| Tropical Andes | IT | -5.8 (-3) | -0.7 (-8) | 4.1 (73) | 0.4 (74) | 191 (88) | 0.6 | 81 (7.1) |

| | | | | | | | | |
|---|---|---|---|---|---|---|---|---|
| | OT2 | -19.3 (-4) | -1.2 (-8) | 2.8 (23) | 0.3 (10) | 437 (77) | 0.3 | 19 (2) |
| | OT3 | -30.4 (-3) | -4 (-7) | 14.1 (40) | 1.6 (25) | 1149 (81) | 0.4 | 43 (5.2) |
| Dry Andes | DA1 | -5.2 (-2) | -0.4 (-4) | 1.8 (62) | 0.2 (106) | 218 (93) | 0.3 | 50 (11.9) |
| | DA2 | -7.4 (-1) | -2 (-4) | 11.3 (59) | 0.1 (14) | 770 (76) | 0.3 | -269 (-32) |
| | DA3 | -32.6 (-5) | -3 (-8) | 8.5 (23) | -0.1 (-3) | 613 (97) | 0.3 | -629 (-27.2) |
| Wet Andes | WA1 | -7 (-3) | -1.1 (-8) | 1.7 (6) | -1.6 (-13) | 237 (93) | 0.3 | -937 (-18.3) |
| | WA2 | -41.2 (-3) | -11.6 (-13) | 10.7 (6) | -11.7 (-9) | 1550 (91) | 0.4 | -454 (-8) |
| | WA3 | -4.9 (-1) | -3 (-7) | 4.4 (14) | 1.1 (5) | 469 (4) | 0.5 | -161 (-4.4) |
| | WA4 | -72 (-2) | -21.4 (-9) | 15.3 (8) | 4.4 (5) | 3746 (57) | 0.4 | -96 (-5.1) |
| | WA5 | 4.5 (1) | -0.3 (-1) | 4.1 (14) | 2.1 (7) | 378 (15) | 0.4 | -407 (-6.5) |
| | WA6 | -23.9 (-2) | -10.5 (-8) | 7.7 (7) | -4.5 (-5) | 1524 (32) | 0.3 | -382 (-10) |

### 3.4 Hydro-glaciological behavior at the catchment scale during the period 2000-2019

In this Section, we focus on three Andean catchments: La Paz (16°S, Tropical Andes), Maipo (33°S, Dry Andes) and Baker (47°S, Wet Andes) (see locations in Figure 2 or 4), where previous glaciological observations and simulations of glacier evolution and water production have been carried out, and *in situ* records are also available. Detailed results for each of the 786 catchments and glaciers included are available in the dataset provided in the Supplementary Material.

### 3.4.1 Glaciological variations in the selected catchments: La Paz (16°S), Maipo (33°S) and Baker (47°S)

Figure 5 shows the annual specific mass balance in the three catchments (2000-2019). The mean over the study period is negative, and there is a negative trend for the annual values toward 2019. For instance, for the Maipo catchment, we estimate a mean annual mass balance of -0.29 ± 0.14 m w.e. yr$^{-1}$, a slightly more negative balance in the Baker catchment (-0.47 ± 0.19 m w.e. yr$^{-1}$), whereas the glaciers in the La Paz catchment show a greater loss of -0.56 ± 0.19 m w.e. yr$^{-1}$. In addition, when considering the annual mass balance values, it is possible to note differences between the catchments. The La Paz catchment shows mostly negative annual mass balance values over the whole period, while in the Baker and Maipo catchments the mass balances are predominantly negative after 2004 and 2009, respectively. Considering the total area and volume changes per catchment in the periods 2000-2009 and 2010-2019, an overall reduction is observed in each of the three catchments. For the La Paz catchment, considering 86% (14 km$^2$) of glacierized area in 2000 (mean glacierized elevation of 5,019 m a.s.l.) and 20 glaciers, the glacierized surface area and volume decrease by -7% (-1 km$^2$) and -11% (-0.1 km$^3$), respectively. For the Maipo catchment, with a larger percentage of simulated glacierized surface area in 2000

(99%, with mean elevation of 4,259 m a.s.l.) and a greater number of glaciers (n = 225), the area and volume
decrease by -1% (-4.2 km$^2$) and -5% (-1 km$^3$), respectively. For the Baker catchment, which contains the largest
glacierized surface area of the three catchments in 2000, we simulated 66% of this glacierized area (1514 km$^2$,
with mean elevation of 1,612 m a.s.l.) and 1805 glaciers: this area shrank by approximately -2% (-36.7 km$^2$),
losing close to -11% (-9.3 km$^3$) of its volume. These results are summarized in Table 3.

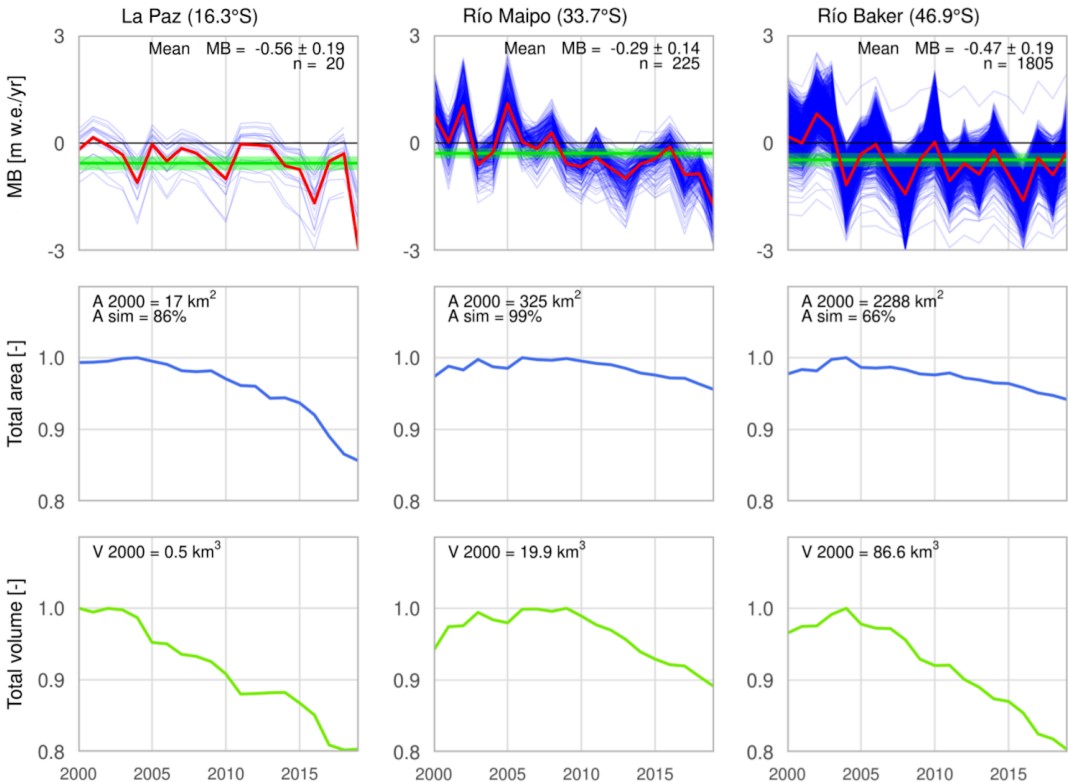

**Figure 5. Recent annual specific mass balance, surface area, and volume variations in the La Paz, Maipo, and Baker catchments from 2000 to 2019. The first row shows the mass balance for each simulated glacier (blue line), as well as the weighted mean mass balance per catchment (red line). The mean geodetic mass balance and its error for the period 2000-2019 are also presented (green bar). The second row presents the total glacierized area per catchment (blue line). The total area from RGI v6.0 and the simulated area percentage are also presented. The last row exhibits the total volume per catchment (green line). The surface area and volume have both been normalized to make it easier to compare the evolution between the catchments.**

**3.4.2 Hydrological contribution of glaciers in the selected catchments: La Paz (16°S), Maipo (33°S) and Baker (47°S)**

The La Paz, Maipo, and Baker catchments display large climatic and glaciological differences over the period
2000-2019. For instance, contrasting cumulative precipitation amounts can be found between the Baker and La
Paz catchments (2224 ± 443 mm yr$^{-1}$ and 791 ± 100 mm yr$^{-1}$, respectively), while the La Paz and Maipo
catchments present the maximum difference in mean annual temperature (1.4 ± 0.5°C and -4.1 ± 0.5°C,
respectively) (Figure 6). At a seasonal scale, precipitation in the Maipo and Baker catchments is concentrated in
autumn and winter (April-September), even if the latter catchment also has a significant amount of precipitation
in summer. Conversely, precipitation in the La Paz catchment mainly occurs in spring and summer (October to
March). In addition, the La Paz and Baker catchments are characterized by the warmest temperatures (>0°C) in
spring and summer; the warmest temperatures for the Maipo catchment occur in summer. Variations in the
climatic conditions are observed between 2000-2009 and 2010-2019. For instance, a decrease in cumulative
precipitation is observed in the Maipo (-30%, -454 mm yr[-1]) and Baker catchments (-2%, -52 mm yr[-1]), but an
increase can be seen in the La Paz catchment (4%, 30 mm yr[-1]). The mean annual temperature increases in the
three catchments (+0.5°C in La Paz and Baker, +0.4°C in Maipo).
The glacier runoff simulation, which considers the glacier melt (ice and snow melt) and rainfall on glaciers
(liquid precipitation), shows strong differences between the catchments (Figure 6). Over the period 2000-2019,
the glaciers in the Baker catchment have the highest mean annual glacier melt (94 ± 19.6 m$^3$/s), followed by
those in the Maipo (15.1 ± 4.2 m$^3$/s) and La Paz catchments (0.5 ± 0.2 m$^3$/s). The rainfall on glaciers contributes
30% to glacier runoff in the Baker catchment (41 ± 10.1 m$^3$/s); a lower value is found in the La Paz catchment
with 17% (0.1 m$^3$/s) followed by the Maipo catchment with 5% (0.8 ± 0.3 m$^3$/s), which is the lowest contribution
of rainfall on glaciers in these catchments. The simulations of glacier runoff changes between the periods
2000-2009 and 2010-2019 for the three catchments show an increase in glacier melt and rainfall on glaciers. The
largest relative increase in mean annual glacier melt is observed in the Maipo with 37% (4.7 m$^3$/s), followed by
the La Paz with 21% (0.09 m$^3$/s) and the Baker catchments with 10% (9 m$^3$/s). Meanwhile, the largest relative
increase in the mean annual rainfall on glaciers is observed in the La Paz catchment (15%, 0.01 m$^3$/s), followed
by the Baker catchment (11%, 4.3 m$^3$/s) and lastly the Maipo catchment (2%, 0.02 m$^3$/s). The results for the
glacier melt and rainfall on glaciers are summarized in Table 3.

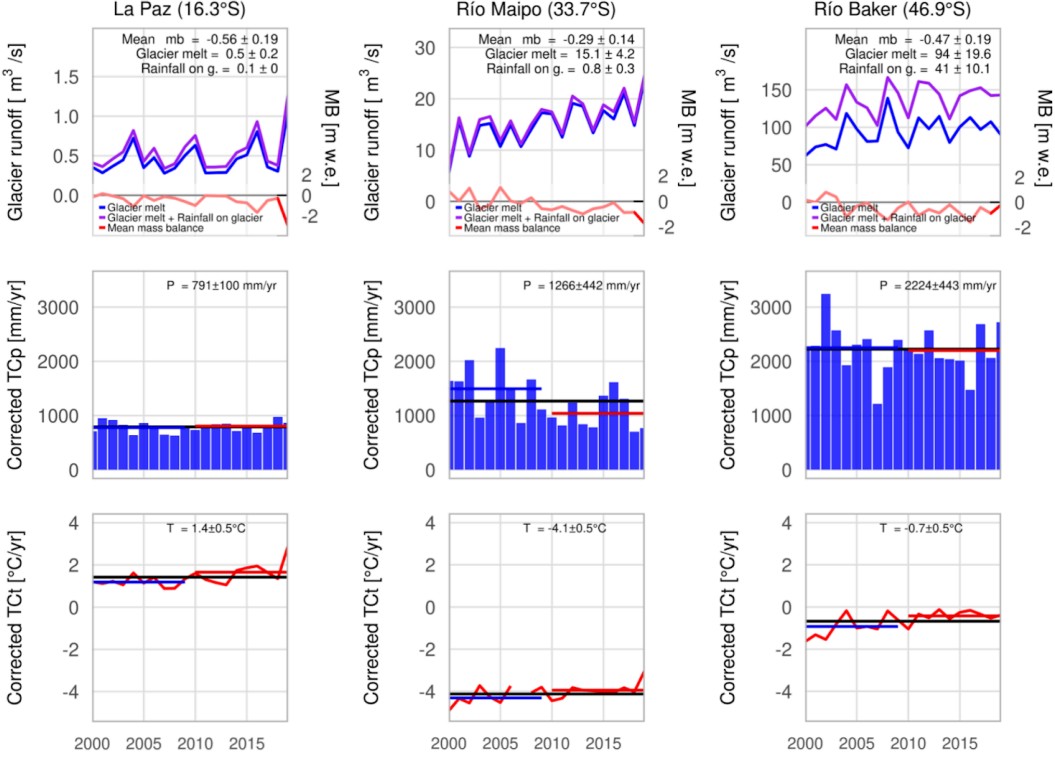

**Figure 6. Hydro-glaciological responses and climate variations in the La Paz, Maipo and Baker catchments from 2000 to 2019. The first row presents the mean annual glacier runoff (purple line = ice melt+snow**

**melt+rainfall on glaciers), the mean annual glacier melt (blue line = ice melt+snow melt), and the annual specific mass balance (red line). The other rows show the mean total annual precipitation and mean annual temperature with the mean annual amount for the periods 2000-2019 (black line), 2000-2009 (blue line) and 2010-2019 (red line).**


In Figure 7, at a mean monthly temporal scale for the period 2000-2019, the glacier melt simulation presents a
short maximum during summer (January-February) in the Maipo and Baker catchments. In contrast, peaks in the
La Paz catchment are extended during spring and summer (November-March) highlighting the so-called
transition season (between September and November) where there is a low amount of rainfall on glaciers and
glacier melt progressively increases. In the Baker catchment, melting begins earlier in September while in Maipo
it begins later (November). The interannual variability of glacier melt over the periods 2000-2009 and
2010-2019 shows a larger contribution from the glacier in the period 2010-2019 for the Maipo catchment.
Furthermore, the simulated rainfall on glaciers is larger mainly during the summer season in all catchments, with
more rainfall in the La Paz catchment (December to February) after the transition season.

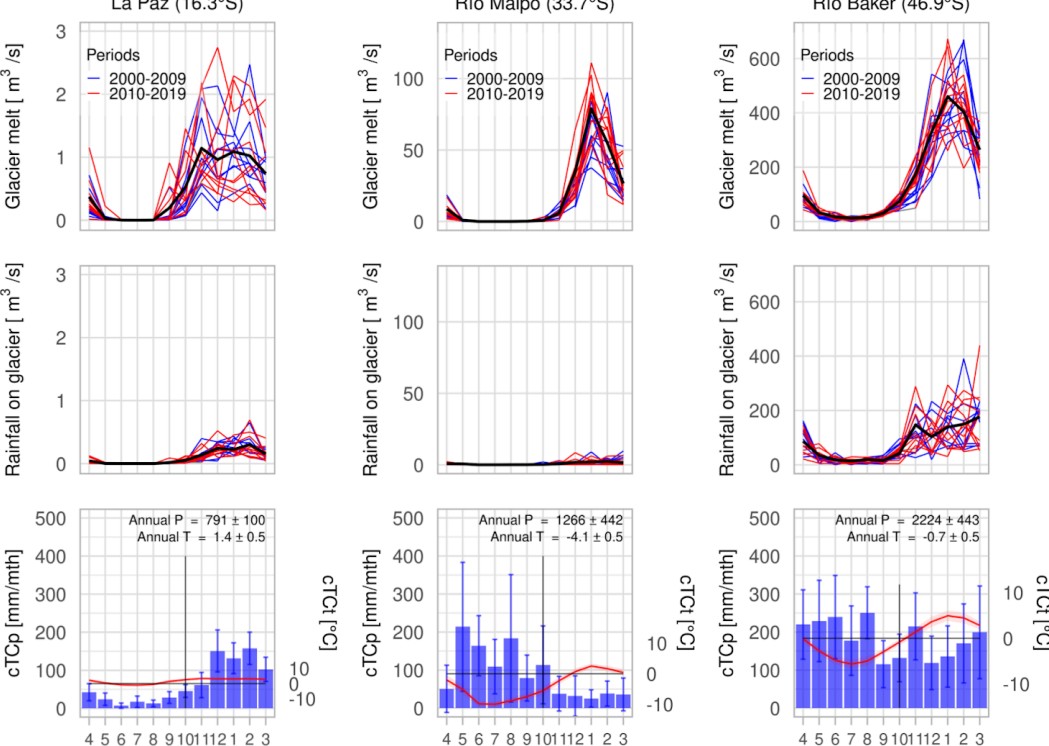

**Figure 7. Monthly hydro-glaciological responses and climate variations in the La Paz, Maipo and Baker catchments from 2000 to 2019. The first and second rows present the mean monthly glacier melt and rainfall on glaciers (black line) and the mean amounts per year during the periods 2000-2009 (red lines) and 2010-2019 (blue lines). In the last row, the climographs show the mean monthly precipitation (blue bars) and temperature (red line) for the period 2000-2019.**


For the mean annual discharge measurements in each catchment and the mean annual simulated glacier runoff
(glacier melt and rainfall on glaciers) between 2000-2019 (Figure 8), we estimate that the largest glacier runoff
contribution is in the Baker catchment (24%), followed by the La Paz (22%) and Maipo catchments (14%),
where all catchments present a similar proportion of glacierized surface area (7.5% to 8.2%). If we consider the
summer season only (January to March), the glacier runoff contribution is highest in the Baker catchment (43%),
followed by the Maipo (36%) and La Paz catchments (18%), where the larger percentage of glacier melt is found
in the Maipo catchment (34%) and the larger percentage of rainfall on glaciers is displayed in the Baker
catchment (12%). Unlike the Maipo and Baker catchments, which present a maximum glacier runoff
contribution in the summer season, the La Paz catchment shows the largest glacier runoff contribution (45%) in
the transition season (September to November).

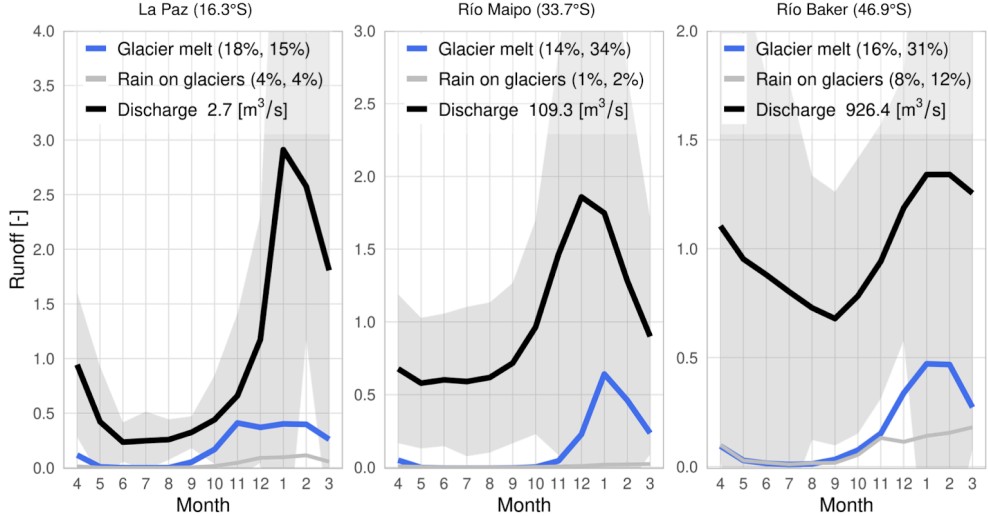

**Figure 8. Monthly simulated glacier runoff (glacier melt + rainfall on glaciers) and discharge measurements in the La Paz, Maipo, and Baker catchments from 2000 to 2019. The results for the glacier melt (blue lines) and rainfall on glacier calculations (gray) are presented, as well as the discharge measurement (black line) and its standard deviation (gray area). The mean annual glacier runoff contribution (as a percentage) and the mean glacier runoff contribution (as a percentage) from January to March are shown in parentheses. The values are normalized by the mean river discharge.**


| Table 3. Hydro-glaciological changes and variations in climate between the periods 2000-2009 and 2010-2019 for the three selected catchments | | | | | | | |
|---|---|---|---|---|---|---|---|
| Region | Catchment | Change in surface area [km²] (%) | Change in volume [km³] (%) | Contribution of the annual glacier melt [m³/s] (%) | Contribution of the annual rainfall on glaciers [m³/s] (%) | Total simulated glacierized area [km²] (%) | cTCt variation [°C] | cTCp variation [mm yr⁻¹] (%) |
| TA | La Paz | -0.96 (-6.7) | -0.1 (-11.5) | 0.09 (21.3) | 0.01 (15.3) | 14.4 (86) | 0.5 | 30 (4) |
| DA | Maipo | -4.2 (-1.3) | -1 (-5) | 4.7 (37) | 0.02 (2.2) | 353.9 (99) | 0.4 | -454 (-30) |
| WA | Baker | -36.7 (-2.4) | -9.3 (-10.7) | 9 (10) | 4.3 (11.2) | 1514 (66) | 0.5 | -52 (-2) |


## 4 Discussion

### 4.1 Comparison with previous studies across the Andes

Huss and Hock (2018) studied 12 Andean catchments across the Andes (1980-2000 and 2010-2030) and estimated an increase in glacier runoff in the Tropical Andes (Santa and Titicaca catchments) and the Dry Andes (Rapel and Colorado catchments). Our results are consistent with these estimates. We show an increase in glacier melt by 40% and 36% in both regions, respectively, between the periods 2000-2009 and 2010-2019. However, in the Wet Andes, Huss and Hock (2018) did not estimate any changes in glacier runoff on the western side of the Andes (Biobio catchment), and instead found a decrease (Río Negro catchment) and an increase (Río Santa Cruz catchment) in glacier runoff on the eastern side of the Andes. Our results for this region show an increase in glacier melt by 8% and a decrease in rainfall on glaciers by -3%.

Based on local reports in the Tropical Andes, the catchment associated with the Los Crespos glacier (catchment id = 6090223080) on the Antisana volcano shows a small decrease in the glacier area of -1% between the periods 2000-2009 and 2010-2019, which is in agreement with Basantes-Serrano et al. (2022). Their study estimated that almost half of the glacier area (G1b, G2-3, G8, G9 and G17) had a positive mass balance during the period 1998-2009 with the largest glacier presenting a mass balance of $0.36 \pm 0.57$ m w.e. $yr^{-1}$, in agreement with our mass balance estimation at the catchment scale of $0.2 \pm 0.5$ m w.e. $yr^{-1}$ (2000-2009). However, in this region, the corrected TerraClimate temperature cannot reproduce the magnitude of the monthly temperature variation (see Figure S2). This limits the effectiveness of the parameter values used in the model to accurately simulate the melting onset and the amount of solid/liquid precipitation. Furthermore, the mass balance simulation is performed through the temperature-index model which does not take the sublimation process into account; and in addition, it runs at a monthly time step thereby limiting the relevant processes that occur hourly. On the other hand, the catchments that contain the Zongo glacier (catchment id = 6090629570) and the Charquini glacier (catchment id = 6090641570) display results that are consistent with the observations (Rabatel et al., 2012; Seehaus et al., 2020; Autin et al. 2022). In addition, our simulated mass balance evaluation on the Zongo glacier shows a low bias (-0.2 m w.e. $yr^{-1}$) with regard to the observations. In the Dry Andes, the catchments associated with the Pascua Lama area (catchment id = 6090836550 and catchment id = 6090840860), the Tapado glacier (catchment id = 6090853340) and the glaciers of the Olivares catchment (catchment id = 6090889690) show consistent results in terms glaciological variations in comparison with the observations (Rabatel et al., 2011; Malmros et al., 2016; Farías-Barahona et at., 2020; Robson et al., 2022). In the Wet Andes, the catchments associated with the Chilean side of the Monte Tronador (catchment id = 6090945100) and the Martial Este and Alvear glaciers in Tierra del Fuego (catchment id = 6090037770) show results that are consistent with previous reports (Rabassa 2010; Ruiz et al., 2017). Despite this, it is possible that our methodology could overestimate precipitation in some catchments; for example, the cumulative precipitation associated with the Nevados de Chillán catchment (catchment id = 6090916140) was estimated at 4023 mm $yr^{-1}$.

At the glaciological region scale, previous studies have reported a large decrease in the percentage of glacier area in the Tropical Andes by -29% (2000-2016) (Seehaus et al., 2019; 2020), followed by the Dry Andes between -29 and -30% (Rabatel et al., 2011; Malmros et al., 2016) although for a longer time-period. In the Wet Andes, Meier et al. (2018) reported a -9% decrease in the glacier area (1986–2016). Our simulations are consistent with these observed glacier area reductions. In addition, Caro et al. (2021) estimated a similar trend across the Andes between 1980-2019 (Tropical Andes = -41%, Dry Andes = -39%, Wet Andes = -24%). On the other hand, we

found high correlations between the mean annual climatic variables and annual mass balance. In the Dry Andes, this correlation was high with precipitation (r = 0.8 ± 0.1, p-value < 0.05) and in the Wet Andes, temperature was correlated with mass balance (r = -0.7 ± 0.1, p-value < 0.05) as previously observed by Caro et al. (2021). These correlations between precipitation or temperature with the annual mass balances for each catchment across the Andes can be reviewed in Table S7 and Figure S5 of the Supplementary Materials.

**4.2. Comparison of our results with previous studies in the three selected catchments**

In the La Paz catchment, Soruco et al. (2015) evaluated the mass balance of 70 glaciers (1997-2006) and their contribution to the hydrological regime. In the present study, we simulated a less negative mass balance (-0.56 ± 0.19 m w.e. yr$^{-1}$ *vs.* -1 m w.e. yr$^{-1}$) considering a larger glacierized area due to the use of RGI v6.0 (with 14.1 km$^2$ in comparison to 8.3 km$^2$). Our estimation of the mean annual glacier runoff (22%) is larger than the previous estimation close to 15% (Soruco et al., 2015). This may be due to the fact that we have considered a warmer 2010-2019 period than the one observed in Soruco et al. (2015). Unlike the previous report, we estimated a larger glacier runoff contribution during the wet season (26%, October to March) and increasing in the transition season (45%, September to November). This increase in glacier runoff contribution given by the model agrees with the larger glacier mass loss observed by Sicart et al. (2007) and Autin et al. (2022) during this season. In the Maipo catchment, we identified a slightly smaller glacierized area (325 km$^2$ for the year 2000, -14%) compared with Ayala et al. (2020) because they considered rock glaciers from the Chilean glacier inventory. In addition, we observed a more negative mass balance after 2008, coinciding with the mega-drought period characterized by a decrease in precipitation and an increase in temperature (Garreaud et al., 2017). The hydrological response to this negative mass balance trend is an increase in glacier runoff since 2000 that is concentrated between December and March. The modeled mean annual glacier runoff contribution estimation is close to 15%, reaching 36% in summer (January-March), is close to Ayala et al. (2020) estimation (16% at the annual scale for the period 1955-2016). However, this comparison between our results and previous studies in the Maipo and La Paz catchments is limited due to the utilization of different inputs, spatial resolutions, time steps, and workflow in the simulated processes where some processes as mass balance of all glaciers was not done. Lastly, in the Baker catchment, Dussaillant et al. (2012) stated that catchments associated with the Northern Patagonian Icefield (NPI) are strongly conditioned by glacier melting. In this respect, Huss and Hock (2018) did not identify glacier runoff changes between the periods 1980-2000 and 2010-2030, they only considered 183 km$^2$ of the glacierized area (-12% until 2020), whereas we estimated a 10% and 11% increase in glacier melt and rainfall on glaciers, respectively, taking a larger glacierized area (1514 km$^2$; -2% until 2020) into account. The relevance of the rainfall on glaciers with regards to the glacier runoff estimated here is close to 30% (including glaciers from east of NPI to the east) which is confirmed by Krogh et al. (2014), who estimated that over 68% of the total precipitation at the catchment scale in east NPI (León and Delta catchments) corresponds to rainfall.

**4.3 Simulation limitations**

Limitations in the simulations result from different sources: (1) the quality/accuracy of the input data; (2) the calibration of the precipitation and melt factors; and (3) the model itself, including its structure and the processes that are not represented. Regarding the evaluation of the corrected TerraClimate temperature using meteorological observations in the Tropical Andes, the corrected TerraClimate data do not reproduce either the

low monthly temperature or the higher temperature in specific months which have a mean bias of 2.1°C (*e.g.* Llan_Up-2 9°S, Zongo at glacier station 16°S). These differences found in the corrected TerraClimate data limit the capacity of the ice/snow melting module to accurately simulate the months in which melting can occur. To account for this, the values of the thresholds used for the melting onset and for the solid/liquid precipitation phase have been adjusted and are described in the limitations (2). On the contrary, in the Dry Andes and Wet Andes, the corrected TerraClimate temperatures are closer to the *in situ* observations (mean bias = 0.2ºC) and present a reliable monthly distribution. This results in model parameter values that are in better agreement with the values used in other studies. Other limitations come from RGI v6.0 because some glaciers are considered as only one larger glacier. For example, in the Dry Andes (catchment id = 6090889690) two large glaciers, the Olivares Gamma and the Juncal Sur, form one (even larger) glacier. These glaciers could underestimate the simulated change in glacier area, limiting the performance of the volume module which depends on the glacier geometry and bedrock shape.

Furthermore, we applied different precipitation factor values in the Tropical Andes (1), Dry Andes (1.9 to 4) and Wet Andes (2.3 to 4), in order to increase the annual mass balance. These values are in agreement with former studies, for example, similar values were used in the Dry Andes (Masiokas et al., 2016; Burger et al., 2019; Farías-Barahona et al., 2020). Values that are too high could lead to an overestimation of precipitation on some glaciers. However, to confirm that the precipitation factor produces realistic precipitation values, we adjusted the standard deviation of the simulated mass balance to the observed mass balance, a method similar to that proposed in Marzeion et al. (2012) and Maussion et al. (2019). On the other hand, the uncertainty of the calibrated melt factors come from the climate and geodetic mass balance datasets used to run and calibrate the model. Indeed, the melting temperature threshold establishes the onset of melting and influences the number of months in which it occurs. On the other hand, the geodetic mass balance defines the accumulated gain or loss per glacier over the calibration period, which in this case spans 20 years. Based on our evaluation of the corrected TerraClimate temperature and simulated mass balance, we correctly reproduce the seasonal melt distribution, associated with a mean underestimated overall annual mass balance of 185 mm w.e. yr$^{-1}$ which however is correlated with the *in situ* data (r = 0.7). According to Rounce et al. (2020), similar results of glacier surface mass balance could be due to different combinations of model parameters. For instance, a wetter (or dryer) and warmer (or colder) parameter set—where high (or low) precipitation factors are compensated by high (or low) temperature biases—can lead to similar recent glacier mass changes and projections. Conversely, the implications for glacier runoff are likely to be significant for both recent and future simulations. In a wetter (or dryer) and warmer (or colder) scenario, there would be increased (or decreased) precipitation and melt, resulting in larger (or smaller) glacier runoff contribution. To address this, we obtained realistic values for precipitation and temperature based on *in situ* spatially distributed measurements and on our field experiences on monitored Andean glaciers. Furthermore, our evaluation of simulations in the three selected catchments enabled us to estimate glacier runoff amounts in the same order of magnitude as previous reports. However, caution must be exercised when using the calibrated melt factors estimated in the Tropical Andes. This is because the temperature in this region was overestimated by an average of 2.4°C, impacting the calibration of the melt factor values. These values should be lower than those estimated here (see Figure 3).

With regards to the structural limitations of the model, it would be relevant to distinguish between ice and snow melt when simulating the glacier melt with two melt factors. In addition, the sublimation on the glacier surface is

very relevant in some glaciers located in the Tropical Andes and Dry Andes 1 (Rabatel et al., 2011; MacDonnell
et al., 2013). However, the OGGM model does not incorporate these processes in glacier runoff and mass
balance simulations.

**5 Conclusion**

In this study, we present a detailed quantification of the glacio-hydrological evolution across the Andes
(11°N-55°S) over the period 2000-2019 using OGGM. Our simulations rely on a glacier-by-glacier calibration of
the changes in glacier volume. Simulations cover 11,282 km$^2$ of the glacierized surface area across the Andes,
taking out account that calving glaciers (mostly in the Patagonian icefields and Cordillera Darwin) were not
considering because calving is not accounted for in the version of glaciological model used here. The simulations
were performed for the first time employing the same methodological approach, a corrected climate forcing and
parameters calibration at the glaciological zone scale throughout the Andes. Evaluation of our simulation outputs
spanned glacier-specific and catchment-scale, integrating *in situ* observations -which is uncommon in regional
studies. From our results, we can conclude the following:
- In relation to glacier runoff composed by glacier melt and rainfall on glacier at the catchment scale; the
largest percentage of studied Andean catchments encompassing 84% of total (661 catchments)
presented an increase in 12% of the mean annual glacier melt (ice and snowmelt) between the periods
2000-2009 and 2010-2019. These catchments present glaciers with higher elevation, larger size and also
a lower mean annual temperature and higher mean annual precipitation compared with glaciers located
in catchments that showed a decrease in glacier melt in the same period which comprise just 12% of
studied catchments. Additionally, the mean annual rainfall on glacier between the periods 2000-2009
and 2010-2019 exhibited a reduction of -2%.
- Special attention must be directed towards the Tropical and Dry Andes regions, as they exhibited the
most significant percentage increase in glacier runoff between the periods 2000-2009 and 2010-2019,
reaching up to 40% due to glacier melt, and 3% due to increased rainfall on glacier over the past
decade. Specifically, the Dry Andes 1 showcased a remarkable 62% increase, while the Inner Tropic
zone exhibited a 73% rise in glacier runoff in the same periods. Notably, these particular glaciological
zones displayed the smallest absolute quantities of glacier runoff across the entire Andes region. The
Dry Andes 1 zone emerges as the most vulnerable glaciological zone to glacier runoff water scarcity in
the Andes.
- Three catchments (La Paz, Maipo and Baker) located in contrasted climatic and morphometric zones
(glaciological zones) are used to evaluate the simulations. Our results show consistency with previous
studies and *in situ* observations. The larger glacier runoff contributions to the catchment water flows
during the period 2000-2019 are quantified for the Baker (43%) and Maipo (36%) catchments during
the summer season (January-March). On the other hand, the larger glacier runoff contribution to the La
Paz catchment (45%) was estimated during the transition season (September to November).
- The correction of temperature and precipitation data, coupled with parameter calibration conducted at
the glaciological zone scale, enabled obtaining annual estimates of glacier mass balance and runoff
closer to what has been measured in glaciers and some Andean catchments. Highlighting the estimation
of annual temperature lapse rates and variability in glacier mass balance through measurements to

correct climate data across distinct glaciological zones. This improvement not only ensures better alignment with local observations but also establishes a more robust tool for forecasting future glacier runoff patterns in the Andes. This method stands apart from global models by specifically addressing the local climate and parameter values inherent to the Andean region.

Lastly, our results help to improve knowledge about the hydrological responses of glacierized catchments across the Andes through the correction of inputs, calibration by glaciers and validation of our simulations considering different glaciological zones. The implementation of this model during the historical period is a prerequisite for simulating the future evolution of the Andean glaciers.

## Code and data availability

Data per glacier in this study is available at https://doi.org/10.5281/zenodo.7890462

## Supplement link

Supplementary information is available at https://doi.org/10.5194/egusphere-2023-888-supplement

## Author contributions

AC, TC and AR were involved in the study design. AC wrote the model implementation and produced the figures, tables and first draft of the manuscript. NC contributed to the model implementation. AC and NG carried out the data curation and TerraClimate temperature evaluation. AC performed the first level of analysis, which was improved by input from TC, AR, NC, FS. All authors contributed to the review and editing of the manuscript.

## Acknowledgments

We acknowledge LabEx OSUG@2020 (Investissement d'Avenir, ANR10 LABX56). The first author would like to thank Dr. Shelley MacDonell (University of Canterbury - CEAZA), Ashley Apey (Geoestudios), Dr. Marius Schaefer (U. Austral de Chile), and Dr. Claudio Bravo and Sebastián Cisternas (CECs, Centro de Estudios Científicos de Valdivia) for the data provided. In addition, the first author thanks the OGGM support team, especially Patrick Schmitt, Dr. Lilian Schuster, Larissa van der Laan, Anouk Vlug, Dr. Rodrigo Aguayo and Dr. Fabien Maussion. Lastly, the first author greatly appreciates discussing the results for specific glaciers or catchments with Dr. Ezequiel Toum (IANIGLA, Argentina), Dr. Álvaro Ayala (CEAZA, Chile), Dr. Lucas Ruiz (IANIGLA, Argentina), Dr. Gabriella Collao (IGE, Univ. Grenoble Alpes, France), Dr. Diego Cusicanqui (IGE-ISTerre, Univ. Grenoble Alpes, France), and Dr. David Farías-Barahona (FAU, U. de Concepción, Chile). All authors are grateful for the comments provided by the two anonymous reviewers and by the editor, which helped considerably improve the scientific quality of this article.

## Financial support

This study was conducted as part of the International Joint Laboratory GREAT-ICE, a joint initiative of the IRD, universities/institutions in Bolivia, Peru, Ecuador and Colombia, and the IRN-ANDES-C2H. This research was funded by the National Agency for Research and Development (ANID)/Scholarship Program/DOCTORADO BECAS CHILE/2019-72200174.

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

Methods and Progress. The International Archives of thePhotogrammetry, Remote Sensing and Spatial
Information Sciences XLI-B4, 125–128. (20), 2016.

Devenish, C., and Gianella, C. Sustainable Mountain Development in the Andes. 20 Years of Sustainable
Mountain Development in the Andes - from Rio 1992 to 2012 and beyond. Lima, Peru: CONDESAN, 2012.

DGA, datos de estudios hidroglaciológicos de Chile [data set], https://snia.mop.gob.cl/BNAConsultas/reportes,
761 2022.


Dussaillant A., Buytaert W., Meier C. and Espinoza F. Hydrological regime of remote catchments with extreme
gradients under accelerated change: the Baker basin in Patagonia. Hydrological Sciences Journal,Volume 57,
https://doi.org/10.1080/02626667.2012.726993, 2012.
Dussaillant, I., Berthier, E., Brun, F., Masiokas, M., Hugonnet, R., Favier, V., et al. Two Decades of Glacier Mass
Loss along the Andes. Nat. Geosci. 12 (10), 802–808. https://doi.org/10.1038/s41561-019-0432-5, 2019.
Farías-Barahona, D., Wilson, R., Bravo, C., Vivero, S., Caro, A., Shaw, T. E., et al. A Near 90-year Record of the
Evolution of El Morado Glacier and its Proglacial lake, Central Chilean Andes. J. Glaciol, 66, 846–860.
https://doi.org/10.1017/jog.2020.52, 2020.
Farinotti, D., Huss, M., Fürst, J. J., Landmann, J., Machguth, H., Maussion, F., et al. A consensus estimate for
the    ice    thickness    distribution    of    all    glaciers    on    Earth.    Nat.    Geosci.    12,    168–173.
https://doi.org/10.1038/s41561-019-0300-3, 2019.
Farinotti, D., Brinkerhoff, D. J., Clarke, G. K., Fürst, J. J., Frey, H., Gantayat, P. & Andreassen, L. M. How
accurate are estimates of glacier ice thickness? Results from ITMIX, the Ice Thickness Models Intercomparison
eXperiment. The Cryosphere, 11(2), 949-970, https://doi.org/10.5194/tc-11-949-2017, 2017.
Favier, V., Wagnon, P., Chazarin, J.-P., Maisincho, L., and Coudrain, A. One-year measurements of surface heat
budget  on  the  ablation  zone  of  Antizana  glacier  15,  Ecuadorian  Andes,  J.  Geophys.  Res.,  109,  D18105,
https://doi.org/10.1029/2003JD004359, 2004.
Fukami, H. and Naruse, R. Ablation of ice and heat balance on Soler glacier, Patagonia. Bull. Glacier Res. 4,
37–42, 1987.
Gao L., Bernhardt M. and Schulz K. Elevation correction of ERA-Interim temperature data in complex terrain.
Hydrol. Earth. Syst. Sci. 16(12): 4661–4673, https://doi.org/10.5194/hess-16-4661-2012, 2012.
Garreaud, R. D., Alvarez-Garreton, C., Barichivich, J., Boisier, J. P., Christie, D., Galleguillos, M., LeQuesne,
C., McPhee, J., and Zambrano-Bigiarini, M.: The 2010–2015 megadrought in central Chile: impacts on regional
hydroclimate       and       vegetation,       Hydrol.       Earth       Syst.       Sci.,       21,       6307–6327,
https://doi.org/10.5194/hess-21-6307-2017, 2017.
Gascoin, S., Kinnard, C., Ponce, R., Lhermitte, S., MacDonell, S., and Rabatel, A.: Glacier contribution to
streamflow  in  two  headwaters  of  the  Huasco  River,  Dry  Andes  of  Chile,  The  Cryosphere,  5,  1099–1113,
https://doi.org/10.5194/tc-5-1099-2011, 2011.
GLACIOCLIM, Données météorologiques [data set], https://glacioclim.osug.fr/Donnees-des-Andes, 2022.
Guido, Z., McIntosh, J. C., Papuga, S. A., and Meixner, T. Seasonal Glacial Meltwater Contributions to Surface
Water in the Bolivian Andes: A Case Study Using Environmental Tracers. J. Hydrol. Reg. Stud. 8, 260–273.
https://doi.org/10.1016/j.ejrh.2016.10.002, 2016.

Hernández, J., Mazzorana, B., Loriaux, T., and Iribarren, P. Reconstrucción de caudales en la Cuenca Alta del
Río Huasco, utilizando el modelo Cold Regional Hydrological Model (CRHM), AAGG2021, 2021.

Hock, R. Temperature index melt modelling in mountain areas. Journal of Hydrology, 282(1-4), 104-115.
https://doi.org/10.1016/S0022-1694(03)00257-9, 2003.

Hugonnet, R., McNabb, R., Berthier, E. et al. Accelerated global glacier mass loss in the early twenty-first
century. Nature 592, 726–731. https://doi.org/10.1038/s41586-021-03436-z, 2021.

Huss, M. and Hock, R. A new model for global glacier change and sea-level rise, Front. Earth Sci., 3, 54,
https://doi.org/10.3389/feart.2015.00054, 2015.

Huss, M. and Hock, R. Global-scale hydrological response to future glacier mass loss, Nature Climate Change,
8, 135–140, https://doi.org/10.1038/s41558-017-0049-x, 2018.

IANIGLA, datos meteorológicos [data set], https://observatorioandino.com/estaciones/, 2022.

Kienholz, C., Rich, J. L., Arendt, A. A., and Hock, R.: A new method for deriving glacier centerlines applied to
glaciers in Alaska and northwest Canada, The Cryosphere, 8, 503–519, https://doi.org/10.5194/tc-8-503-2014,
826 2014.


Koizumi, K. and Naruse R. Measurements of meteorological conditions and ablation at Tyndall Glacier,
Southern Patagonia, in December 1990. Bulletin of Glacier Research, 10, 79-82, 1992.

Krögh, S.A., Pomeroy, J.W., McPhee, J. Physically based hydrological modelling using reanalysis data in
Patagonia. J. Hydrometeorol. http://dx.doi.org/10.1175/JHM-D-13-0178.1, 2014.

Lehner B, Verdin K, Jarvis A. Hydrological data and maps based on Shuttle elevation derivatives at multiple
scales (HydroSHEDS)-Technical Documentation, World Wildlife Fund US, Washington, DC, Available at
http://hydrosheds.cr.usgs.gov, 2016.

MacDonell, S., Kinnard, C., Mölg, T., Nicholson, L., and Abermann, J. Meteorological drivers of ablation
processes on a cold glacier in the semi-arid Andes of Chile, The Cryosphere, 7, 1513–1526,
https://doi.org/10.5194/tc-7-1513-2013, 2013.

Malmros, J. K., Mernild, S. H., Wilson, R., Yde, J. C., and Fensholt, R. Glacier Area Changes in the central
Chilean and Argentinean Andes 1955-2013/14. J. Glaciol. 62, 391–401. https://doi.org/10.1017/jog.2016.43,
844 2016.

Marangunic C., Ugalde F., Apey A., Armendáriz I., Bustamante M. and Peralta C. Ecosistemas de montaña de la
cuenca alta del río Mapocho, Glaciares en la cuenca alta del río Mapocho: variaciones y características
principales. AngloAmerican - CAPES UC, 2021.
Mark, B. and Seltzer, G. Tropical glacier meltwater contribution to stream discharge: A case study in the
Cordillera Blanca, Peru. J. Glaciol, 49(165), 271-281. https://doi.org/10.3189/172756503781830746, 2003.
Marzeion, B., Jarosch, A. H., and Hofer, M.: Past and future sea-level change from the surface mass balance of
glaciers, The Cryosphere, 6, 1295–1322, https://doi.org/10.5194/tc-6-1295-2012, 2012.
Masiokas, M. H., Christie, D. A., Le Quesne, C., Pitte, P., Ruiz, L., Villalba, R., et al. Reconstructing the Annual
Mass Balance of the Echaurren Norte Glacier (Central Andes, 33.5° S) Using Local and Regional Hydroclimatic
Data. The Cryosphere 10 (2), 927–940. https://doi.org/10.5194/tc-10-927-2016, 2016.
Masiokas, M. H., Rabatel, A., Rivera, A., Ruiz, L., Pitte, P., Ceballos, J. L., et al. A Review of the Current State
and Recent Changes of the Andean Cryosphere. Front. Earth Sci. 8 (6), 1–27.
https://doi.org/10.3389/feart.2020.00099, 2020.
Mateo, E. I., Mark, B. G., Hellström, R. Å., Baraer, M., McKenzie, J. M., Condom, T., Rapre, A. C., Gonzales,
G., Gómez, J. Q., and Encarnación, R. C. C.: High-temporal-resolution hydrometeorological data collected in the
tropical Cordillera Blanca, Peru (2004–2020), Earth Syst. Sci. Data, 14, 2865–2882,
https://doi.org/10.5194/essd-14-2865-2022, 2022.
Maussion, F., Butenko, A., Champollion, N., Dusch, M., Eis, J., Fourteau, K., et al. The open global glacier
model (OGGM) v1.1. Geoscientific Model. Develop. 12, 909–931. https://doi.org/10.5194/gmd-12-909-2019,
871 2019.

Meier, W.J-H., Grießinger, J., Hochreuther, P. and Braun, M.H. An Updated Multi-Temporal Glacier Inventory
for the Patagonian Andes With Changes Between the Little Ice Age and 2016, Front. Earth Sci., 6:62.
https://doi.org/10.3389/feart.2018.00062, 2018.
Millan, R., Mouginot, J., Rabatel, A. et al. Ice velocity and thickness of the world's glaciers. Nat. Geosci. 15,
124–129. https://https://doi.org/10.1038/s41561-021-00885-z, 2022.
NASA JPL. NASADEM Merged DEM Global 1 arc second V001 [Data set]. NASA EOSDIS Land Processes
DAAC. https://doi.org/10.5067/MEaSUREs/NASADEM/NASADEM_HGT.001, 2020.

Rabassa, J. El cambio climático global en la Patagonia desde el viaje de Charles Darwin hasta nuestros días. Revista de la Asociación Geológica Argentina, 67(1), 139-156, 2010.

Rabatel, A., Bermejo, A., Loarte, E., Soruco, A., Gomez, J., Leonardini, G., Vincent, C., and Sicart, J. E.: Relationship between snowline altitude, equilibrium-line altitude and mass balance on outer tropical glaciers: Glaciar Zongo – Bolivia, 16∘ S and Glaciar Artesonraju – Peru, 9∘ S, J. Glaciol., 58, 1027–1036, https://doi.org/10.3189/2012JoG12J027, 2012.

Rabatel, A., Castebrunet, H., Favier, V., Nicholson, L., and Kinnard, C. Glacier Changes in the Pascua-Lama Region, Chilean Andes (29° S): Recent Mass Balance and 50 Yr Surface Area Variations. The Cryosphere 5 (4), 1029–1041. https://doi.org/10.5194/tc-5-1029-2011, 2011.

Rabatel, A., Francou, B., Soruco, A., Gomez, J., Cáceres, B., Ceballos, J. L., et al. Current State of Glaciers in the Tropical Andes: A Multi-century Perspective on Glacier Evolution and Climate Change. The Cryosphere 7, 81–102. https://doi.org/10.5194/tc-7-81-2013, 2013.

Ragettli, S., and Pellicciotti, F. Calibration of a Physically Based, Spatially Distributed Hydrological Model in a Glacierized basin: On the Use of Knowledge from Glaciometeorological Processes to Constrain Model Parameters. Water Resour. Res. 48 (3), 1–20. https://doi.org/10.1029/2011WR010559, 2012.

RGI Consortium. Randolph Glacier Inventory - A Dataset of Global Glacier Outlines, Version 6. Boulder, Colorado USA. NSIDC: National Snow and Ice Data Center, https://doi.org/10.7265/4m1f-gd79, 2017.

Rivera, A. Mass balance investigations at Glaciar Chico, Southern Patagonia Icefield, Chile. PhD thesis, University of Bristol, UK, 303 pp, 2004.

Robson, B. A., MacDonell, S., Ayala, Á., Bolch, T., Nielsen, P. R., and Vivero, S. Glacier and rock glacier changes since the 1950s in the La Laguna catchment, Chile, The Cryosphere, 16, 647–665, https://doi.org/10.5194/tc-16-647-2022, 2022.

Ruiz, L., Berthier, E., Viale, M., Pitte, P., and Masiokas, M. H. Recent geodetic mass balance of Monte Tronador glaciers, northern Patagonian Andes, The Cryosphere, 11, 619–634, https://doi.org/10.5194/tc-11-619-2017, 2017.

Rounce, D.R., Khurana, T., Short, M.B., Hock, R., Shean, D.E., Brinkerhoff, D.J. Quantifying parameter uncertainty in a large-scale glacier evolution model using Bayesian inference: application to High Mountain Asia. Journal of Glaciology, 66(256):175-187, https://doi.org/10.1017/jog.2019.91, 2020.

Schaefer M., Rodriguez J., Scheiter M., and Casassa, G. Climate and surface mass balance of Mocho Glacier, Chilean Lake District, 40°S. Journal of Glaciology, 63(238), 218-228, https://doi.org/10.1017/jog.2016.129, 2017.

Schuster. L., Rounce, D.R., Maussion, F. Glacier projections sensitivity to temperature-index model choices and calibration strategies. Annals of Glaciology, 1-16, https://doi.org/10.1017/aog.2023.57, 2023

Seehaus, T., Malz, P., Sommer, C., Lippl, S., Cochachin, A., and Braun, M. Changes of the tropical glaciers throughout Peru between 2000 and 2016 – mass balance and area fluctuations, The Cryosphere, 13, 2537–2556, https://doi.org/10.5194/tc-13-2537-2019, 2019.

Seehaus, T., Malz, P., Sommer, C., Soruco, A., Rabatel, A., and Braun, M. Mass balance and area changes of glaciers in the Cordillera Real and Tres Cruces, Bolivia, between 2000 and 2016. J. Glaciol, 66(255), 124-136. https://doi.org/10.1017/jog.2019.94, 2020.

SENAMHI, datos hidrometeorológicos de Perú [data set], https://www.senamhi.gob.pe/?&p=descarga-datos-hidrometeorologicos, 2022.

Shaw, T. E., Caro, A., Mendoza, P., Ayala, Á., Pellicciotti, F., Gascoin, S., et al. The Utility of Optical Satellite Winter Snow Depths for Initializing a Glacio-Hydrological Model of a High-Elevation, Andean Catchment. Water Resour. Res. 56 (8), 1–19. https://doi.org/10.1029/2020WR027188, 2020.

Sicart, J. E., Wagnon, P., and Ribstein, P. Atmospheric controls of heat balance of Zongo Glacier (16°S. Bolivia). J. Geophys. Res. 110:D12106. https://doi.org/10.1029/2004JD005732, 2005.

Sicart, J. E., P. Ribstein, B. Francou, B. Pouyaud, and T. Condom. Glacier mass balance of tropical Zongo Glacier, Bolivia, comparing hydrological and glaciological methods, Global Planet. Change, 59(1), 27– 36, https://doi.org/10.1016/j.gloplacha.2006.11.024, 2007.

Sicart, J. E., R. Hock, and D. Six. Glacier melt, air temperature, and energy balance in different climates: The Bolivian Tropics, the French Alps, and northern Sweden, J. Geophys. Res., 113, D24113, https://doi.org/10.1029/2008JD010406, 2008.

Stuefer, M. Investigations on Mass Balance and Dynamics of Moreno Glacier Based on Field Measurements and Satellite Imagery. Ph.D. Dissertation, University of Innsbruck, Innsbruck, 1999.

Stuefer, M., Rott, H. and Skvarca, P. Glaciar Perito Moreno, Patagonia: climate sensitivities and glacier characteristics preceding the 2003/04 and 2005/06 damming events. J. Glaciol., 53 (180), 3–16. https://doi.org/10.3189/172756507781833848, 2007.

Soruco, A., Vincent, C., Rabatel, A., Francou, B., Thibert, E., Sicart, J. E., et al. Contribution of Glacier Runoff
to Water Resources of La Paz City, Bolivia (16° S). Ann. Glaciol. 56 (70), 147–154.
https://doi.org/10.3189/2015AoG70A001, 2015.

Takeuchi, Y., Naruse R. and Satow K. Characteristics of heat balance and ablation on Moreno and Tyndall
glaciers, Patagonia, in the summer 1993/94. Bulletin of Glacier Research, 13, 45-56, 1995.

WGMS. Global Glacier Change Bulletin No. 4 (2018-2019). Michael Zemp, Samuel U. Nussbaumer, Isabelle
Gärtner-Roer, Jacqueline Bannwart, Frank Paul, and Martin Hoelzle (eds.), ISC (WDS) / IUGG (IACS) / UNEP /
UNESCO / WMO, World Glacier Monitoring Service, Zurich, Switzerland, 278 pp. Based on database version
https://doi.org/10.5904/wgms-fog-2021-05, 2021.

Zhang, G.Q., Bolch, T., Yao, T.D., Rounce, D.R., Chen, W.F., Veh, G., King, O., Allen, S.K., Wang, M.M.,
Wang, W.C. Underestimated mass loss from lake-terminating glaciers in the greater Himalaya. Nat. Geosci. 16,
333–338. https://doi.org/10.1038/s41561-023-01150-1, 2023.

Zimmer, A., Meneses, R. I., Rabatel, A., Soruco, A., Dangles, O., and Anthelme, F. Time Lag between Glacial
Retreat and Upward Migration Alters Tropical alpine Communities. Perspect. Plant Ecol. Evol. Syst. 30, 89–102.
https://doi.org/10.1016/j.ppees.2017.05.003, 2018.