# Peer review of "Hydrological Response of Andean Catchments to Recent Glacier Mass Loss"

_EGUsphere, 2023_

## Referee Comment (RC1)

**Review of "Hydrological response of Andean catchments to recent glacier mass loss"**
**by Caro et al.**

This study investigates the changes in glacier mass, area, and runoff for different glaciated catchments in the Southern Andes from 2000-2019. The study uses the Open Global Glacier Model, calibrated with geodetic mass balance data from 2000-2019, and forced by a bias-corrected climate dataset. The focus of the study is on all land-terminating glaciers (i.e., lake- and marine-terminating glaciers are excluded) and specific attention is given to how the changes in climate over 2000-2009 vs. 2010-2019 affect the glacier runoff across the different catchments, which notably span different climatologies. The main conclusions are that most glaciers are losing mass leading to increases in runoff in the Tropical Andes and Dry Andes. Furthermore, results are consistent with previous studies and the glacier contribution to runoff is highest for some catchments in the summer and others in the transition season prior to summer.

While I was excited to read this study, and believe it's an important topic(!), I found the writing to be challenging to follow. The readability of the study would be greatly improved by providing more details and context to many aspects of the methods and results. For example, values were often reported as a change, but it was unclear what the change was related to. Explicit definitions of the glacier runoff and how contributions are calculated would also help. This might also clarify potential issues with comparisons to other studies where different variables or different time periods appeared to be compared. The study also provides a lot of model results for various catchments and various parameters such that paragraphs of the results almost read as a catalogue of the changes with even more details provided in the supplementary material. Making the results a bit more concise and highlighting the key aspects may improve the readability and highlight the novel aspects of the study as well.

I believe the challenges associated with my ability to understand the methods and results greatly affected my review and made it hard for me to understand the novelty of the study and the major gap that this fills in the literature or our advanced understanding of the hydrological response. Contextualizing the results in terms of the impacts (e.g., is a change in runoff a good or bad thing for downstream water availability?) or showing the added value of the bias-corrected climate data may also help highlight some of the novelty. For example, my understanding is the model focused on 2000-2019 because that's when the climate data was available as well as in-situ observations. However, a 20-year time period is fairly short for evaluating the impacts of climate change; therefore, I'm left wondering what is the added value of focusing on this time period? Does it lay the foundation for an improved model to model historic or project future changes? Does it improve the predictive capabilities of seasonal runoff?

Given the lack of novelty and major issues with respect to readability, I believe the manuscript requires major revisions and another round of review. I believe the topic of understanding the hydrological response of glaciers to climate change, especially in an important area like the Southern Andes, is important and warrants publication; however, significant improvements are required and it's unclear to me if the scope of these revisions would be suitable for a typical response to reviewers or if a reject and resubmit would be better. See additional comments below.

General Comments
I found that a good portion of the text lacks context or details that are needed to understand what the authors are stating. For example, the abstract provides good overall numbers, but lacks context such that if the reader begins by reading the abstract it is difficult to understand what the numbers of referring to. Similar challenges in reading the text due to a lack of context were found throughout the text. See specific comments.

I also found the description of the methods to be quite confusing. Figure 1, for example, doesn't have an arrow going from the corrected data to forcing the simulations. The caption also makes it sound like these are two separate workflows (one input data to run the model and another to perform a correction); however, it's unclear to me how these are done separately if the mass balance model is then calibrated after these are performed? I highly recommend modifying the methods section to make things clear. See specific comments below.

More details on how glacier runoff is defined (e.g., fixed vs. moving gauge) and what glacier contribution to runoff means (is this the ratio of the glacier melt to the glacier melt plus precipitation just at the outlet of the glaciers?) are needed. This would help provide context to interpret the results.

The novelty of the study is unclear. For example, the conclusion that most glacierized catchments are losing glacier mass is already known from the observations used to calibrate the model. The third conclusion is that the results are consistent with previous studies. It would be useful for the novel aspects of this study to be included.

Specific Comments
L20 – a bit unclear what "reduced in 93% of the catchments between the periods 2000-2009 and 2010-2019" means. Is this meaning that 7% of the glaciers grew? Is this by area? by number?

L21 – -9% and +0.4 degC compared to what? 2010-2019 vs. 2000-2009? Or compared to a prior year?

L23 – consider adding "calibrated" prior to "melt factors" to make this clear that it's a model parameter.

L25 – specify "contribution" to what? Total runoff in the catchments? Glacier melt vs. glacier runoff?

L27 – "a high mean correlation" between what? Annual mass balance measurements? Discharge?

L29 – "increases" compared to what?

L33 – I assume "largest ice concentration" is referring to glacier area or glacier volume; however, there are many glacierized areas with more area and volume (e.g., Alaska, Arctic

Canada, High Mountain Asia) (Millan et al. 2022). Hence, is this the glacier area divided by catchment area? Please clarify what is meant here as this currently does not appear to be correct.

L34 – Consider changing "provide the water supply" to "supply water"

L44 – Suggest defining what "glacier contribution" refers to. I assume fraction of glacier runoff relative to the total catchment runoff?

L47-48 & L54-55 – There appears to be an implicit changing of definitions of glacier runoff, since the area loss being compensated by the increase mass loss suggests a "moving-gauge" runoff framework (i.e., calculating runoff at the outlet of the glacier as it moves over time), while Huss and Hock (2018) use a "fixed-gauge" runoff framework where they include snow melt and rainfall of off-glacier areas in their comparisons.

L99-103 – "the elevation difference between a glacier inside a TerraClimate grid and the mean TerraClimate grid elevation" is unclear.  What part of the glacier was used?  The min, mean, max elevation?  Or I assume this was used to adjust the temperature for every elevation bin?  The $cTC_t$ appears to refer to both the mean glacier elevation and the glacier catchment, which doesn't make sense. Please be specific on how variables are defined and calculated.

L113 – Suggest stating how these were selected.  Might be just continuing the sentence with the following criteria; however, unclear as to why 10 glaciers were selected and this analysis wasn't performed over all 18 glaciers?  Did something go wrong or is there a lack of confidence in the modeling or data from some glaciers?

L118 – "using a similar methodology as that for …" : this is just a pearson correlation, right?  I would recommend stating that to improve readability.  I also don't think it saves space by saying similar methodology when the methodology is only a few words.

Equation 2 – is there a reason this comes after the paragraph instead of in line like equation 1?

L119 – Check journal standards, but normally section comes first, i.e., "Section 2.2"

L133-134 – what was quantified?  This statement is very vague.  I suggest improving readability by being explicitly.  For example, "The OGGM toolbox was used to derive the glacier-wide mass balance from the elevation change estimates from Hugonnet et al. (2021)." is very clear about what the Hugonnet product is, what "quantified" refers to, and what processing was done by the toolbox.  Otherwise, Hugonnet et al. (2021) quantified mass changes, so very unclear what was actually done.

L150 – consider lake- and marine-terminating as I don't think "fresh terminating" makes sense and often see tide as tidewater glaciers, not tide-terminating.

L152 – state OGGM version here. Also consider changing to "was not included in the version of OGGM used in this study" or something similar since OGGM does have a calving parameterization that could be used.

L154 – what are contradictory variations?

L156 – selected for what? "to represent glaciological regions with different climatic and morphometric characteristics"? Sentence doesn't make sense as currently written.

L158-159 – if previous hydro-glaciological studies have already quantified the impact, then what is the novelty of this study? Suggest adding something regarding limitations of previous studies here or how the proposed work will be an advance.

L170 – "short description of the" seems unnecessary. Consider just "OGGM" or "OGGM details".

L172 – "contains enough default input data" is incredibly vague. Do you mean provides pre-processed datasets such as DEMs, glacier hypsometry, glacier flowlines, etc. that can be used to explicitly simulate …?

L174 – Previous text states that the study is done by glacier, not glacier catchment. Results are aggregated to glacier catchments, but as the text currently reads it sounds like the model is running at the glacier catchment.

L176 – Suggest deleting "it is possible" and replacing with what was done.

L178 – How is the geodetic mass balance rate calibrated? Isn't this input data used for calibration?

L180-182 – Strikes me as odd that the ice thickness inversion for OGGM cites the ice thickness model intercomparison as opposed to Maussion et al. (2019; GMD)

L188 – "where" should be lowercase?

L188 – I found this description of the precipitation/snow difficult to understand. Was the solid precipitation really scaled by the precipitation factor? If so, isn't this problematic since the TerraClimate data is at a different elevation (previously described) and thus the snow/rain differentiation will be off? Thus, the precipitation factor would be also accounting for the fact that the amount of precipitation that falls as snow increases at higher elevations due to colder climates as well as any biases in the precipitation datasets itself? Please clarify.

L190-192 – Again, I'm confused by how this temperature threshold is performed since it states that 100% of precipitation is classified as snow between 0-2.1 degC and then 0% between 2-4.1 degC. First, the bounds don't match (i.e., one goes to 2.1 and the other starts at 2). Second, make this clear that this is a model parameter (realized this from Table 1) with supposedly a 2.1 degree range and the linear interpolation varies over the span of two degrees? Perhaps explicitly state the model parameters in a separate sentence too. The way it's currently written is hard to understand even for someone quite familiar with OGGM.

Table 1 – bottom appears to be cutoff

L199 – Great! Include this information earlier as its clear to understand.

L203-205 – Perhaps I'm misunderstanding the term "model output", but normally I think of model output as a result and thus the result is not calibrated. Rather, the model parameters are calibrated. It's also normally easier to understand "[insert model parameter] is calibrated such that [insert model and observations used]". Please clarify.

L211 – I'm not suggesting redoing the study for something this small, but why wasn't the mass balance output monthly or the model timestep switched from calendar to hydrological years such that the comparison is done properly?

L214 – Suggest referencing this workflow figure earlier in the description where it would be useful to understand the workflow as opposed to it being referenced for the first time once the description is complete.

Figure 1 – I may be misunderstanding the figure, but "observed and simulated MB" do not appear to come from the Corrected climate data. This is rather confusing as my understanding was that the corrected climate data was used to run the MB simulations, no? Glacier melt and rainfall on glacier also appear to be floating in the figure: I believe these are meant to define runoff, but suggest putting them in the runoff box; otherwise, they don't make much sense.

L226 – typically climate change is referring to long-term changes. Here, climate change is being used to refer to changes between two decades. Couldn't differences at this short of time scales be due to interannual climate variability (AMOC, El Nino, etc.)?

L237 – what is meant by amplitude? Isn't it just the mean monthly temperature?

L248-252 – This language is very confusing to follow. Why was only 36% of the glacierized area simulated and not all of the glaciers here, which I thought is what the methods suggested? Furthermore, why does this differ so much from the 85% and 79% stated next? Or is this just meant to state the fraction of the total glacier area in the whole region? If the latter, then this should go in the Methods.

L258 – delete "negative" as the value is shown as a negative number.

L261 – specific wording comment, but Figure 2b technically shows the specific mass balance, not the volume.

Figure 3 – how is the mode shown if these are floating values? Or was the calibration performed in a step-wise fashion? It strikes me as odd that the modes are always a minimum or maximum value indicating the bounds are often what the melt factor reaches for calibration.

L290 – if mentioning sublimation, may be worth noting that sublimation is "implicitly" included in the model, since the observed mass loss is including sublimation while the modeled mass loss

does not.  That means that the model parameters are implicitly being calibrated to account for this process.

L293-294 – why not include this where the simulations are initially mentioned in L278, "To test our results …"

Figure 4 – if using an acronym (e.g., G Melt) need to state what it stands for in the caption (even if obvious).

L421-426 – see previous not about comparing fixed-gauge runoff (Huss and Hock 2018) to moving-gauge runoff (this study – this is my assumption as its not particularly clear what happens as the glacier retreats).  This information should be provided.  If comparing two different ways of estimating runoff, then the comparison isn't really meaningful.

L440 – at first use, it may be useful to mention what this id refers to since it's clearly not RGI which the inventory stated was used.

L466 – typo? "considering a largest glacierized area"?  "larger" perhaps?

L464-469 – This suggests that mass balances over two different time periods are being compared?  If that's true, this should definitely not be done as the climate forcing may be completely different.  Why not model the same time periods (which OGGM can do) and thus be able to do a proper comparison?

L478 – "is" close to?

L479-L481 – sentence does not make sense.  Please clarify/rephrase what is being stated.

L492 – temperature index "model" …

L497 – how were all of these differences taken into account? The present study appears to only perform a single set of simulations with one set of model parameters for each glacier.  Am I missing something?

L502 – higher "melt" factor values?

L502 & L512-514 – The problem being described seems identical to the overparameterization problem associated with glacier evolution models (e.g., Rounce et al. 2020; JoG). Hence, it might be worth mentioning that this might be what is occurring, i.e., changes in temperature (essentially a temperature bias correction performed using in-situ measurements) are being compensated by changes in the degree-day factor, and thus caution should be taken to avoid overinterpreting calibrated model parameters (especially given that they'll compensate for other factors not accounted for in the model such as changing surface conditions, avalanching, etc. as mentioned in the limitations section). Changing the melt temperature threshold and the rain/snow temperature thresholds by 2 degrees for the Tropical Andes compared to the Dry/Wet Andes

(Table 1) is essentially the exact same thing as using a temperature bias value of 2 degrees (i.e., an additive value that adjusts the temperature by 2 degrees).

L530 – how did the geodetic mass balance define the maximum melting per glacier?  Again, this may stem from my misunderstanding of what's actually being calibrated and how the calibration was performed.

L532 – what does a "true seasonal melting distribution" mean?  Is this the observations?

L551-554 – Are these percentage increase in the Tropical Andes and Dry Andes referring to the same Inner Tropic and Dry Andes 1 zones or is this statement meant to refer to different zones? Please clarify as this is unclear.

---

## Author Comment (AC1)

**Review of "Hydrological response of Andean catchments to recent glacier mass loss"**
**by Caro et al.**

Dear reviewer, in this document we present our replies (in purple) together with the changes we made to answer your comments. For each of your comments (in *Italics-black*), the original draft text is written in red color, whereas our proposed changes to the original draft text and comments are in blue color.

*This study investigates the changes in glacier mass, area, and runoff for different glaciated catchments in the Southern Andes from 2000-2019. The study uses the Open Global Glacier Model, calibrated with geodetic mass balance data from 2000-2019, and forced by a bias-corrected climate dataset. The focus of the study is on all land-terminating glaciers (i.e., lake- and marine-terminating glaciers are excluded) and specific attention is given to how the changes in climate over 2000-2009 vs. 2010-2019 affect the glacier runoff across the different catchments, which notably span different climatologies. The main conclusions are that most glaciers are losing mass leading to increases in runoff in the Tropical Andes and Dry Andes. Furthermore, results are consistent with previous studies and the glacier contribution to runoff is highest for some catchments in the summer and others in the transition season prior to summer. While I was excited to read this study, and believe it's an important topic(!), I found the writing to be challenging to follow. The readability of the study would be greatly improved by providing more details and context to many aspects of the methods and results. For example, values were often reported as a change, but it was unclear what the change was related to. Explicit definitions of the glacier runoff and how contributions are calculated would also help. This might also clarify potential issues with comparisons to other studies where different variables or different time periods appeared to be compared. The study also provides a lot of model results for various catchments and various parameters such that paragraphs of the results almost read as a catalogue of the changes with even more details provided in the supplementary material. Making the results a bit more concise and highlighting the key aspects may improve the readability and highlight the novel aspects of the study as well.*
*I believe the challenges associated with my ability to understand the methods and results greatly affected my review and made it hard for me to understand the novelty of the study and the major gap that this fills in the literature or our advanced understanding of the hydrological response. Contextualizing the results in terms of the impacts (e.g., is a change in runoff a good or bad thing for downstream water availability?) or showing the added value of the*

*bias-corrected climate data may also help highlight some of the novelty. For example, my understanding is the model focused on 2000-2019 because that's when the climate data was available as well as in-situ observations. However, a 20-year time period is fairly short for evaluating the impacts of climate change; therefore, I'm left wondering what is the added value of focusing on this time period?*

*Does it lay the foundation for an improved model to model historic or project future changes?*

*Does it improve the predictive capabilities of seasonal runoff?*

*Given the lack of novelty and major issues with respect to readability, I believe the manuscript requires major revisions and another round of review. I believe the topic of understanding the hydrological response of glaciers to climate change, especially in an important area like the Southern Andes, is important and warrants publication; however, significant improvements are required and it's unclear to me if the scope of these revisions would be suitable for a typical response to reviewers or if a reject and resubmit would be better. See additional comments below.*

**Reply:**

We acknowledge reviewer 1 for his/her careful reading on the manuscript. We appreciate his/her interest in our study and the positive overall comments. The criticisms highlighted in this introductory comment regarding the focus/novelty of the study have been considered.

**General Comments**

*I found that a good portion of the text lacks context or details that are needed to understand what the authors are stating. For example, the abstract provides good overall numbers, but lacks context such that if the reader begins by reading the abstract it is difficult to understand what the numbers of referring to. Similar challenges in reading the text due to a lack of context were found throughout the text. See specific comments.*

**Reply:**

Many sections of the manuscript have been rewritten considering your general comments and the more specific ones hereafter. Details are given in the replies to the specific comments and rewritten sections of the manuscript are pasted at the end of this document.

*I also found the description of the methods to be quite confusing. Figure 1, for example, doesn't have an arrow going from the corrected data to forcing the simulations. The caption also makes it sound like these are two separate workflows (one input data to run the model and another to*

*perform a correction); however, it's unclear to me how these are done separately if the mass balance model is then calibrated after these are performed? I highly recommend modifying the methods section to make things clear. See specific comments below.*

*More details on how glacier runoff is defined (e.g., fixed vs. moving gauge) and what glacier contribution to runoff means (is this the ratio of the glacier melt to the glacier melt plus precipitation just at the outlet of the glaciers?) are needed. This would help provide context to interpret the results.*

**Reply:**

The required details have been provided and Fig. 1 has been corrected (see new figure below). More details concerning the description of this figure are given in the specific comment L214. Here, you can see the new Fig. 1.

[Figure]

*The novelty of the study is unclear. For example, the conclusion that most glacierized catchments are losing glacier mass is already known from the observations used to calibrate the model. The third conclusion is that the results are consistent with previous studies. It would be useful for the novel aspects of this study to be included.*

**Reply:**

The novelty of the study has been more clearly stated: 1) Application of the same approach all along the Andes; 2) region-specific calibration of the parameters of the model. In other words, this study proposes a regionalization of the parameters in the OGGM model. We have edited

the abstract, Introduction, and conclusion to keep the coherence. You can see these changes as answers to your specific comments, as well as, at the end of this document.

**Specific Comments**

*L20 – a bit unclear what "reduced in 93% of the catchments between the periods 2000-2009 and 2010-2019" means. Is this meaning that 7% of the glaciers grew? Is this by area? by number?*

Original text:

"Our results show that the glacier volume (-8.3%) and surface area (-2.2%) are reduced in 93% of the catchments between the periods 2000-2009 and 2010-2019."

**Reply:**

The sentence has been rewritten to avoid ambiguity:

"Our results at the Andes scale show that the glacier volume and surface area were reduced by 8.3% and 2.2%, respectively, between the periods 2000-2009 and 2010-2019."

*L21 – -9% and +0.4 degC compared to what? 2010-2019 vs. 2000-2009? Or compared to a prior year?*

Original text:

"Our results show that the glacier volume (-8.3%) and surface area (-2.2%) are reduced in 93% of the catchments between the periods 2000-2009 and 2010-2019. This glacier loss is associated with changes in climate conditions (precipitation = -9%; temperature = +0.4 ± 0.1°C) inducing an increase in the mean annual glacier melt of 12% (86.5 m$^3$/s) and a decrease in the mean annual rainfall on glaciers of -2% (-7.6 m$^3$/s).

Edited text:

"Our results at the Andes scale show that the glacier volume and surface area were reduced by 8.3% and 2.2%, respectively, between the periods 2000-2009 and 2010-2019. The glacier loss during these periods is associated with a decrease in precipitation (9%) and an increase in temperature (+0.4 ± 0.1°C). Between the two periods (2000-2009 and 2010-2019) glacier and climate variations have led to a 12% increase in mean annual glacier melt (86.5 m$^3$/s) and a decrease in mean annual rainfall on glaciers of -2% (-7.6 m$^3$/s) across the Andes, both variables compose the glacier runoff."

*L23 – consider adding "calibrated" prior to "melt factors" to make this clear that it's a model Parameter.*

Original text:

"We find a regional pattern in the melt factors showing decreasing values from the Tropical Andes toward the Wet Andes"

Reply: We reformulated the abstract, editing these lines as follows.

Edited text:

"The related calibrated parameters, such as melt factor (for mass balance) and Glen A (for ice thickness), show strong alignment with cold/warm and dry/wet environmental conditions."

*L25 – specify "contribution" to what? Total runoff in the catchments? Glacier melt vs. glacier Runoff?*

Original text:

"A negative mass balance trend is estimated in the three documented catchments (glacierized surface area > 8%), showing the largest mean glacier contribution during the transition season (September-November) in La Paz (Bolivia) (45%) followed by Baker (Chile) (43%) and Maipo (Chile) (36%) during the summer season (January-March)."

Edited text:

"The catchment scale results indicate comparable glacier runoff contribution with previous studies in the Maipo catchment (34°S, Chile). During the transition season, we suggest a larger glacier runoff contribution in the La Paz catchment (16°S, Bolivia). Additionally, we calculated for the first time the glacier runoff contribution in the Baker catchment (47°S, Chile)."

*L27 – "a high mean correlation" between what? Annual mass balance measurements? Discharge?*

Original text:

"In addition, our evaluation in the monitored glaciers indicates an underestimation of the mean simulated mass balance by 185 mm w.e. yr$^{-1}$ and a high mean correlation (r = 0.7)."

Reply: We removed these lines.

*L29 – "increases" compared to what?*

Original text:

"We conclude that the large increases in the simulated glacier melt in the Dry Andes (36%) and the Tropical Andes (24%) have helped to improve our knowledge of the hydro-glaciological characteristics at a much wider scale than previous studies, which focused more on a few select catchments in the Andes."

Reply: We reformulated the conclusion in these lines as follows.

Edited text:

"In summary, this calibrated and validated model, organized by glaciological zones and grounded in our local understanding, utilizing the same methodological approach, stands as a crucial requirement for simulating future glacier runoff in the Andes."

*L33 – I assume "largest ice concentration" is referring to glacier area or glacier volume; however, there are many glacierized areas with more area and volume (e.g., Alaska, Arctic Canada, High Mountain Asia) (Millan et al. 2022). Hence, is this the glacier area divided by catchment area? Please clarify what is meant here as this currently does not appear to be correct.*

Original text:

"The largest ice concentration in the southern hemisphere outside the Antarctic ice sheet is found in the Andes (RGI Consortium, 2017)."

Reply: Thanks for your comment. We incorporated the original reference. Note that we are referring to the Southern Hemisphere

Edited text:

"The largest glacierized area in the southern hemisphere outside the Antarctic ice sheet is found in the Andes (RGI Consortium, 2017; Masiokas et al., 2020)."

*L34 – Consider changing "provide the water supply" to "supply water"*

Original text:

"Andean glaciers provide the water supply for roughly 45% of the population in the Andean countries (Devenish and Gianella, 2012) and for ecosystems (Zimmer et al., 2018; Cauvy-Fraunié and Dangles, 2019)."

Edited text:

"Andean glaciers supply water for roughly 45% of the population in the Andean countries (Devenish and Gianella, 2012) and for ecosystems (Zimmer et al., 2018; Cauvy-Fraunié and Dangles, 2019)."

*L44 – Suggest defining what "glacier contribution" refers to. I assume fraction of glacier runoff relative to the total catchment runoff?*

Original text:

"In the Tropical Andes, the glacier contribution at the annual scale was estimated to be approximately 12% and 15% in the Río Santa (9°S) and La Paz (16°S) catchments, respectively (Mark and Seltzer, 2003; Soruco et al., 2015)."

Edited text:

"Based on local studies, the glacier runoff contribution (glacier runoff relative to the total catchment runoff) in the Tropical Andes was estimated to be around 12% and 15% in the Río Santa (9°S) and La Paz (16°S) catchments, respectively (Mark and Seltzer, 2003; Soruco et al., 2015)."

*L47-48 & L54-55 – There appears to be an implicit changing of definitions of glacier runoff, since the area loss being compensated by the increase mass loss suggests a "moving-gauge" runoff framework (i.e., calculating runoff at the outlet of the glacier as it moves over time), while Huss and Hock (2018) use a "fixed-gauge" runoff framework where they include snow melt and rainfall of off-glacier areas in their comparisons.*

Original text:

L46 to 48

"For the La Paz catchment, Soruco et al. (2015) found no change in the glacier runoff contribution for the period 1997-2006 compared with the longer 1963-2006 period. This was attributed to the fact that the glacier surface reduction over the time-period was compensated by their increasingly negative mass balance"

L54 to 55

"Despite this, Hock and Huss (2018) did not identify changes in the glacier runoff of the Baker catchment since 1980-2000."

Reply: We incorporated the glacier runoff definition used in Hock and Huss (2018) as follows in L41 to 44. Also, we added a new justification in relation to our results.

Edited text:

"For instance, the global-scale study by Huss and Hock (2018) comprised 12 Andean catchments (1980-2100). They defined glacier runoff as all the melt water and rainfall coming from the initially glacierized area as given by the Randolph Glacier Inventory version 4.0. and found an increase in glacier runoff in the Tropical and Dry Andes during the recent decades, but

a more contrasted signal in the Wet Andes: no glacier runoff changes were observed in some catchments, whereas others showed a reduction or an increase. However, their estimations overlook the diverse climates and morphologies of Andean glaciers (Caro et al., 2021). This affects the simulation results, as they heavily rely on climate inputs and calibrated parameters. For instance, varying temperature lapse rates could result in significant disparities in glacier melt and the determination of solid/liquid precipitation on glaciers (Schuster et al., 2023). Furthermore, the selection of precipitation factor values is also crucial."

L99-103 – "the elevation difference between a glacier inside a TerraClimate grid and the mean TerraClimate grid elevation" is unclear. What part of the glacier was used? The min, mean, max elevation? Or I assume this was used to adjust the temperature for every elevation bin? The cTCt appears to refer to both the mean glacier elevation and the glacier catchment, which doesn't make sense. Please be specific on how variables are defined and calculated.

Original text:

"The corrected TerraClimate temperature at the mean glacier elevation or in glacier catchments (cTCt) was calculated using the following equation: Equation 1. where Γ is the temperature LR estimated here, and Δh is the elevation difference between a glacier inside a TerraClimate grid and the mean TerraClimate grid elevation."

Edited text:

"The corrected TerraClimate temperature at the mean elevation of glacier (cTCt) was calculated using the following equation: Equation 1. where Γ is the temperature LR estimated here, and Δh is the elevation difference between a glacier elevation inside a TerraClimate grid and the mean TerraClimate grid elevation."

L113 – Suggest stating how these were selected. Might be just continuing the sentence with the following criteria; however, unclear as to why 10 glaciers were selected and this analysis wasn't performed over all 18 glaciers? Did something go wrong or is there a lack of confidence in the modeling or data from some glaciers?

Original text:

"We ran 31 simulations for 18 glaciers with mass balance measurements across the Andes using Pf values between 1 and 4 taking previous studies into account (Masiokas et al., 2016; Burger et al., 2019; Farías-Barahona et al., 2020). In the end, 10 glaciers were selected (see Table S3)"

Edited text:

"We ran 31 simulations for 18 glaciers with mass balance measurements across the Andes using Pf values between 1 and 4 taking previous studies into account (Masiokas et al., 2016; Burger et al., 2019; Farías-Barahona et al., 2020). Ultimately, 10 glaciers were selected (see Table S3), because their simulated mass balances showed a closer standard deviation in comparison with measurements"

*L118 – "using a similar methodology as that for …" : this is just a pearson correlation, right? I would recommend stating that to improve readability. I also don't think it saves space by saying similar methodology when the methodology is only a few words.*

Equation 2 – is there a reason this comes after the paragraph instead of in line like equation 1?

Original text:

"The simulated annual mass balance was evaluated on 15 monitored glaciers using a similar methodology as that for the cTCt evaluation"

Edited text:

"The simulated annual mass balance was evaluated on 15 monitored glaciers using a Pearson correlation coefficient and bias (as the average difference) from simulated mass balance and measured mass balance"

Reply: We will move the Equation 2 to line 116.

*L119 – Check journal standards, but normally section comes first, i.e., "Section 2.2"*

Reply: Thank you, we checked and corrected these errors in the new version.

*L133-134 – what was quantified? This statement is very vague. I suggest improving readability by being explicitly. For example, "The OGGM toolbox was used to derive the glacier-wide mass balance from the elevation change estimates from Hugonnet et al. (2021)." is very clear about what the Hugonnet product is, what "quantified" refers to, and what processing was done by the toolbox. Otherwise, Hugonnet et al. (2021) quantified mass changes, so very unclear what was actually done.*

Original text:

"Hugonnet et al.'s (2021) product was quantified for each glacier using the OGGM toolbox"

Edited text:

"The Hugonnet et al. (2021) geodetic mass balance is available for each glacier from the OGGM toolbox"

*L150 – consider lake- and marine-terminating as I don't think "fresh terminating" makes sense and often see tide as tidewater glaciers, not tide-terminating.*

Original text:

"Calving glaciers (fresh and tide terminating,"

Edited text:

"Calving glaciers (lake- and marine-terminating,"

*L152 – state OGGM version here. Also consider changing to "was not included in the version of OGGM used in this study" or something similar since OGGM does have a calving parameterization that could be used.*

Original text:

"were not considered as the calving process is not currently implemented in this OGGM version"

Edited text:

"were not considered because the calving process implemented in this version of OGGM (1.5.3) which relies on Hugonnet et al. (2019) data to calibrate the simulated mass balance, could exhibit significant uncertainty when applied to these particular glaciers. In this regard, Zhang et al. (2023) estimated an underestimation of glacier mass loss for lake-terminating glaciers using geodetic methods, accounting for a subaqueous mass loss of $10 \pm 4\%$ in the central Himalaya during the period 2000 to 2020. Their findings revealed that the total mass loss for certain glaciers was underestimated by as much as $65 \pm 43\%$."

*L154 – what are contradictory variations?*

Original text:

"The other remaining 4,514 $km^2$ filtered glacierized surface area corresponds to glacierized catchments that present contradictory variations in terms of glacier volume and surface area"

Edited text:

"The other remaining 4,514 $km^2$ filtered glacierized surface area corresponds to glacierized catchments that present an increase in glacier volume but a reduction in the glacierized surface area"

*L156 – selected for what? "to represent glaciological regions with different climatic and morphometric characteristics"? Sentence doesn't make sense as currently written.*

Original text:

"We selected the La Paz (Soruco et al., 2015), Maipo (Ayala et al., 2020) and Baker (Dussaillant et al., 2012) catchments located in glaciological regions with different climatic and morphometric characteristics (Caro et al., 2021)"

Edited text:

"We selected the La Paz (Soruco et al., 2015), Maipo (Ayala et al., 2020) and Baker (Dussaillant et al., 2012) catchments located in glaciological regions with different climatic and morphometric characteristics (Caro et al., 2021) to evaluate our simulations in terms of glacier changes and glacier runoff contributions over the period 2000-2019"

*L158-159 – if previous hydro-glaciological studies have already quantified the impact, then what is the novelty of this study? Suggest adding something regarding limitations of previous studies here or how the proposed work will be an advance.*

Original text:

"In these catchments, previous hydro-glaciological studies have quantified the impact of glacier changes and its hydrological contribution"

Edited text:

"In the La Paz and Maipo catchments, previous hydro-glaciological studies have quantified the impact of glacier changes and their hydrological contribution. However, these studies often overlook relevant processes such as variations in precipitation, temperature corrections, and the simulation of glacier dynamics. On the other hand, in the Baker catchment, there are currently no estimations of glacier runoff contributions. These three catchments allow us to make comparisons with our regional simulations at the Andes scale using consistent data (e.g., corrected climate datasets and glacier outlines) and methods (e.g., simulating mass balance, dynamics, and glacier runoff) to verify simulation results of the same magnitude, update previous results, and provide new glacier runoff estimates. For example, it is necessary to understand what occurs during the prolonged dry period in Central Chile and Argentina."

*L170 – "short description of the" seems unnecessary. Consider just "OGGM" or "OGGM details".*

Original text:

"2.2 Short description of the OGGM"

Edited text:

"2.2 OGGM details"

*L172 – "contains enough default input data" is incredibly vague. Do you mean provides pre-processed datasets such as DEMs, glacier hypsometry, glacier flowlines, etc. that can be used to explicitly simulate …?*

Original text:

"OGGM is a modular and open-source numerical workflow implemented in Python that contains enough default input data to simulate the glacier mass balance and ice dynamics using calibrated parameter values for each glacier entity individually."

Edited text:

"OGGM is a modular and open-source numerical workflow implemented in Python that provides pre-processed datasets such as DEMs, glacier hypsometry, glacier flowlines, etc. that can be used to explicitly simulate glacier mass balance and ice dynamics using calibrated parameter values for each glacier entity individually. Here, we ran OGGM from Level 2, comprising the flowlines and their downstream lines. However, we used a new baseline climate time series (corrected TerraClimate) as input data. We also calibrated the mass balances and the bed inversion (ice thickness), that allowed us to obtain hydrological outputs (glacier runoff) (details in https://docs.oggm.org/en/v1.4.0/input-data.html)."

*L174 – Previous text states that the study is done by glacier, not glacier catchment. Results are aggregated to glacier catchments, but as the text currently reads it sounds like the model is running at the glacier catchment.*

Original text:

"The spatio-temporal configuration used in this study is the glacierized catchment and the monthly time step."

Edited text:

"The spatio-temporal configuration of the model used in this study is at the glacier scale and at the monthly time step. In a second time, results were analyzed by glacierized catchment, glaciological zone and region."

*L176 – Suggest deleting "it is possible" and replacing with what was done.*

Original text:

"From these input data, it is possible to obtain annual outputs such as the surface mass balance"

Edited text:

"From these input data we computed annual outputs such as the surface mass balance, glacier area and volume"

*L178 – How is the geodetic mass balance rate calibrated? Isn't this input data used for calibration?*

Original text:

"The geodetic mass balance rate and the glacier volume parameters were calibrated."

Reply and edited text: We removed this line but added the lines 203 to 206

"Modeled processes such as the surface mass balance and glacier volume were calibrated (Table 1 and Figure 2). The calibration procedure of the parameters was applied per glacier to match the simulated mass balance 2000-2019 to the geodetic mass balance product from Hugonnet et al. (2021). The simulated glacier volume was calibrated using Farinotti et al. (2019) product at a glaciological zone scale to fit the Glen A parameter. In other words, the same Glen A parameter was used for each glaciological zone"

*L180-182 – Strikes me as odd that the ice thickness inversion for OGGM cites the ice thickness model intercomparison as opposed to Maussion et al. (2019; GMD)*

Reply: Thanks for this comment. In Maussión et al. (2019) is it possible to read: "It is a mass-conservation approach largely inspired by Farinotti et al. (2009), but with distinct characteristics." Because of that, we propose the next correction.

Original text: "Assuming a bed shape, it estimates the ice thickness based on mass conservation and shallow-ice approximation (Farinotti et al., 2017)."

Edited text: "Assuming a bed shape, it estimates the ice thickness based on mass conservation and shallow-ice approximation (Farinotti et al., 2009; Maussion et al., 2019)."

*L188 – "where" should be lowercase?*

Original text: "Where"

Edited text: "where". We corrected these errors in the draft.

*L188 – I found this description of the precipitation/snow difficult to understand. Was the solid precipitation really scaled by the precipitation factor? If so, isn't this problematic since the TerraClimate data is at a different elevation (previously described) and thus the snow/rain differentiation will be off? Thus, the precipitation factor would be also accounting for the fact that the amount of precipitation that falls as snow increases at higher elevations due to colder*

*climates as well as any biases in the precipitation datasets itself? Please clarify.*

Original text: To clarify we edited the line 110

"In addition, the precipitation was scaled (cTCp) using precipitation factors ($Pf$) for each glaciological zone across the Andes"

Please, consider that the precipitation on a glacier comes from the TerraClimate grid and thus has the same distribution across the glacier, where just the precipitation factor increased it. Equation 1 is applied just on temperature.

Edited text:

"In addition, the total precipitation was scaled (cTCp) using precipitation factors ($Pf$) for each glaciological zone across the Andes (see the relationship between solid precipitation and Pf in equation 3)". In a second step we discriminate snow and rain fall using

In line 192 we added the reference

"using a linear regression between these temperature thresholds to obtain the solid/liquid precipitation fraction (Maussion et al., 2019)"

*L190-192 – Again, I'm confused by how this temperature threshold is performed since it states that 100% of precipitation is classified as snow between 0-2.1 degC and then 0% between 2-4.1 degC. First, the bounds don't match (i.e., one goes to 2.1 and the other starts at 2). Second, make this clear that this is a model parameter (realized this from Table 1) with supposedly a 2.1 degree range and the linear interpolation varies over the span of two degrees? Perhaps explicitly state the model parameters in a separate sentence too. The way it's currently written is hard to understand even for someone quite familiar with OGGM.*

*Table 1 – bottom appears to be cutoff*

Original text:

"$T_{melt}$ is the monthly air temperature above which ice melt is assumed to occur (from 0°C to 2.1 °C). $TCp_{snow}$ is calculated as a fraction of the total precipitation ($cTCp$) where 100% is obtained if $cTCt_i <= T_{isnow}$ (between 0-2.1°C) and 0% if $cTCt_i >= T_{irain}$ (between 2-4.1 °C), using a linear regression between these temperature thresholds to obtain the solid/liquid precipitation fraction"

Edited text:

"In addition to Pf and Mf parameters, parameters related to the snow/ice onset ($T_{melt}$) and precipitation fraction ($T_{isnow}$ and $T_{irain}$) are considered. Their values are different across the Andes. $T_{melt}$ is the monthly air temperature above which snow/ice melt is assumed to occur (0°C for the Dry and Wet Andes and 2.1°C for the Tropical Andes). $TCp_{snow}$ is calculated as a

fraction of the total precipitation ($cTCp$) where 100% is obtained if $cTCt\ i$ <= $Tisnow$ (0°C for the Dry and Wet Andes and 2.1°C for the Tropical Andes) and 0% if $cTCt\ i$ >= $Tirain$ (2°C for the Dry and Wet Andes and 4.1°C for the Tropical Andes), using a linear regression between these temperature thresholds to obtain the solid/liquid precipitation fraction"

In Table 1 we added the units in the volume parameter column and we edited the table line.

*L199 – Great! Include this information earlier as its clear to understand.*
Reply: We will move L199-200 to L170.

*L203-205 – Perhaps I'm misunderstanding the term "model output", but normally I think of model output as a result and thus the result is not calibrated. Rather, the model parameters are calibrated. It's also normally easier to understand "[insert model parameter] is calibrated such that [insert model and observations used]". Please clarify.*
Original text:
"Model outputs such as the surface mass balance and glacier volume were calibrated (Table 1 and Figure 2). The calibration procedure was applied per glacier to match the simulated mass balance 2000-2019 to the geodetic mass balance product from Hugonnet et al. (2021)."
Edited text:
"Simulated processes such as the surface mass balance and glacier volume were calibrated (Table 1 and Figure 2). The calibration procedure was applied per glacier to match the simulated mass balance 2000-2019 to the geodetic mass balance product from Hugonnet et al. (2021)."
Reply. In addition, we will correct these inconsistencies throughout the article.

*L211 – I'm not suggesting redoing the study for something this small, but why wasn't the mass balance output monthly or the model timestep switched from calendar to hydrological years such that the comparison is done properly?*
Original text:
"Although the OGGM outputs are in calendar years and the observations are in hydrological years"
Reply: A lower correlation is expected in the comparison between simulated regional output and local observations. Some aspects of our methodology were: regional simulation cannot reproduce local processes (observed in maximum and minimum values) and, measurements of

mass balance using glaciological methods rarely considered the whole glacier because they use point measurements usually in the accessible areas of the ablation zone.

Because of that, we look to evaluate very general similitudes, where the bias (mean differences between simulated and mean mass balance) could be more important than the correlation. In addition, we know that the consideration of different months could generate differences in the bias estimated for the same glacier, but we are looking at what is happening across the Andes. Without a doubt, your improvements must be implemented in future runnings.

Finally, when we ran the algorithms of the OGGM model, most of the tutorials accessible nowadays were not available. And extracting the monthly mass balance was a difficult task among many others more relevant to solve, in our opinion.

*L214 – Suggest referencing this workflow figure earlier in the description where it would be useful to understand the workflow as opposed to it being referenced for the first time once the description is complete.*

*Figure 1 – I may be misunderstanding the figure, but "observed and simulated MB" do not appear to come from the Corrected climate data. This is rather confusing as my understanding was that the corrected climate data was used to run the MB simulations, no? Glacier melt and rainfall on glacier also appear to be floating in the figure: I believe these are meant to define runoff, but suggest putting them in the runoff box; otherwise, they don't make much sense.*

Reply. We cited Figure 1 in the first paragraph of section 2 Data and methods.

Edited text:

"This section comprises the processed data used as input and during the simulation procedure. The simulation workflow is described in Figure 1."

Reply. Also we edited the Figure 1 description.

Original text:

"Figure 1. Workflow per glacier simulation using OGGM between 2000 and 2019. Two groups of input data were used: one to run the model and the second to correct/evaluate the TerraClimate temperature (cTCt) and precipitation (cTCp). Then, the mass balance and glacier volume were calibrated. Lastly, results such as the cTCt and glacier mass balance were evaluated at 34 meteorological stations and on 15 glaciers with mass balance observations."

Edited text:

"Figure 1. Workflow per glacier simulation using OGGM between 2000 and 2019. Two groups of input data were used: one to run the model and the second to correct/evaluate the TerraClimate temperature (cTCt) and precipitation (cTCp). Then, the mass balance and glacier volume were

calibrated. Lastly, results such as the cTCt and glacier mass balance were evaluated at 34 meteorological stations and on 15 glaciers with mass balance observations. The corrections in OGGM and outside box refer to analyses performed by running the model and also analyzing data outside the model tool. An example is the estimation of temperature lapse rates, which were estimated from in situ measurements but introduced in the OGGM model as a parameter value."

We edited the Figure 1 as:

[Figure]

*L226 – typically climate change is referring to long-term changes. Here, climate change is being used to refer to changes between two decades. Couldn't differences at this short of time scales be due to interannual climate variability (AMOC, El Nino, etc.)?*

Original text:

"Meanwhile, the climate change between the periods 2000-2009 and 2010-2019 across the Andes shows a"

Reply: We replazed "climate change" by "variations in climate"

*L237 – what is meant by amplitude? Isn't it just the mean monthly temperature?*

Original text:

"where cTCt cannot represent the mean monthly temperature amplitude"

Edited text:

"where cTCt cannot represent the mean monthly temperature"

*L248-252 – This language is very confusing to follow. Why was only 36% of the glacierized area simulated and not all of the glaciers here, which I thought is what the methods suggested? Furthermore, why does this differ so much from the 85% and 79% stated next? Or is this just meant to state the fraction of the total glacier area in the whole region? If the latter, then this should go in the Methods.*

Original text:

"The annual mass balance and glacier dynamics per glacier are simulated by taking 36% of the total glacierized surface area across the Andes (11°N-55°S) into account to obtain the glacier area and glacier volume at an annual time scale, as well as the glacier runoff (glacier melting and rainfall on glaciers) at a monthly time scale. In more detail, over 85% of the glacierized surface area in the Dry Andes (18°S-37°S) and 79% in the Tropical Andes (11°N-18°S) is considered, which corresponds to 11% (3,377 km2, in 321 catchments) of the total glacierized area of the Andes. For the Wet Andes (37°S-55°S), 29% of the glacierized surface area in the region is considered, which corresponds to 26% (7,905 km2, in 465 catchments) of the total area in the Andes (see the distribution of 255 the catchments in Figure 2a). The simulated lower glacierized surface area in the Wet Andes results from the filtering out of the numerous calving glaciers found there."

Reply. We explained details of the simulated surface area in lines 126-129 and 147-149. In addition, in line 252 we mention the lowest % of the simulated area in the Wet Andes, related mainly to the filtering of calving glaciers. Calving glaciers compose nearly half of the total glacierized surface across the Andes (lines 541-544).

L126-129.

"We used version 6.0 of the Randolph Glacier Inventory (RGI Consortium, 2017) to extract the characteristics of each glacier, e.g., location, area, glacier front in land or water. The RGI v6.0 was checked using the national glacier inventories compiled by Caro et al. (2021), filtering every RGI glacier that was not found in the NGI, to obtain a total glacierized surface area of 30,943 km2 (filtering 633 km2). The glacier extent in the RGI v6.0 is representative of the early 2000s."

L147-149.

"We selected 786 catchments with a surface area between 3,236 and 20 km2 across the Andes (11°N-55°S), including 13,179 glaciers with a total surface area of 11,282 km2 (36% of the total glacierized surface area in the Andes)."

L252

"In more detail, over 85% of the glacierized surface area in the Dry Andes (18°S-37°S) and 79% in the Tropical Andes (11°N-18°S) is considered, which corresponds to 11% (3,377 km2, in 321 catchments) of the total glacierized area of the Andes. For the Wet Andes (37°S-55°S), 29% of the glacierized surface area in the region is considered, which corresponds to 26% (7,905 km2, in 465 catchments) of the total area in the Andes (see the distribution of 255 the catchments in Figure 2a)."

*L258 – delete "negative" as the value is shown as a negative number.*

Original text:

"associated with a negative mean annual mass balance of"

Edited text:

"associated with a mean annual mass balance of"

*L261 – specific wording comment, but Figure 2b technically shows the,*
*not the volume.*

*Figure 3 – how is the mode shown if these are floating values? Or was the calibration performed in a step-wise fashion? It strikes me as odd that the modes are always a minimum or maximum value indicating the bounds are often what the melt factor reaches for calibration.*

Original text:

"The loss in glacier volume (Figure 2b) is largest (-47.8 km3, -9%) in the Wet Andes"

Reply: We used the output called "volume" defined by the OGGM tutorial as the estimated decadal differences (2000-2009 and 2010-2019). Because of that we called it "loss in glacier volume" (https://oggm.org/tutorials/stable/notebooks/beginner/deal_with_errors.html).

In addition, Cogley et al., 2011 defined specific mass balance as "Mass balance expressed per unit area, that is, with dimension [M L–2] or [M L–2 T–1];" and "Specific mass balance may be reported for a point on the surface, a column of unit cross section, or a larger volume such as an entire glacier or a collection of glaciers".

Regarding Figure 3. During the calibration of the snow/ice melt, a melt factor value is fit by glacier considering the Hugonnet et al. (2019) geodetic mass balance in 20 yrs. The mode was

estimated from 13,000 glaciers. You can see an example of melt factor distributions in the next figures for OT3 (mode = 533) and D3 (mode = 89), which show maximum and minimum mode values respectively.

[Figure]

[Figure]

*L290 – if mentioning sublimation, may be worth noting that sublimation is "implicitly" included in the model, since the observed mass loss is including sublimation while the modeled mass loss does not. That means that the model parameters are implicitly being calibrated to account for this process.*

Reply: We add the next lines at the end of line 291. "However, sublimation is implicitly included in the model through the calibrated melt factor values, which are derived from measured mass balance data by Hugonnet et al. (2019). As a result, our estimates of snow/ice melt in the DA1 zone tend to be overestimated."

*L293-294 – why not include this where the simulations are initially mentioned in L278, "To test our results …"*

Repy. We replace the sentence from lines 278-279
Original text:

"To test our results we evaluated the simulated mass balance in 15 monitored glaciers (Tables S4 and S5 and Figure S3)" by the edited sentences from lines 293-294

Edited text:"The simulated mass balance evaluation for the 15 glaciers can be found in Tables S4, S5 and Figures S3 and S4 of the Supplementary Materials."

We also removed the lines  293-294.

*Figure 4 – if using an acronym (e.g., G Melt) need to state what it stands for in the caption (even if obvious).*

Original text:

"Figure 4. Recent glacier runoff components across the Andes. The total glacier melt and rainfall on glaciers are comprised of the mean differences between the periods 2010-2019 and 2000-2009 per catchment (n = 786). (a) It shows the distribution of the glaciological zones (11°N-55°S), followed by (b) glacier melt and (c) rainfall on glaciers at the catchment scale. The (d) total annual glacier melt is presented in each glaciological zone."

Edited text:

"Figure 4. Recent glacier runoff components across the Andes. The total glacier melt and rainfall on glaciers represent the mean differences between the periods 2010-2019 and 2000-2009 per catchment (n = 786). (a) It shows the distribution of the glaciological zones (11°N-55°S), followed by (b) glacier melt and (c) rainfall on glaciers at the catchment scale. The (d) total annual glacier melt is presented in each glaciological zone. G. melt and Rainfall refer to changes in (b) Glacier melt and (c) Rainfall on glaciers, respectively, meanwhile, G. melt in the axis y in (d) refers to cumulative annual glacier melt by glaciological zone."

*L421-426 – see previous not about comparing fixed-gauge runoff (Huss and Hock 2018) to moving-gauge runoff (this study – this is my assumption as its not particularly clear what happens as the glacier retreats). This information should be provided. If comparing two different ways of estimating runoff, then the comparison isn't really meaningful.*

Original text:

"Hock and Huss (2018) studied 11 Andean catchments across the Andes (1980-2000 and 2010-2030) and estimated an increase in glacier runoff in the Tropical Andes (Santa and Titicaca catchments) and the Dry Andes (Rapel and Colorado catchments). Our results are consistent with these estimates. We show an increase in glacier melt by 40% and 36% in both regions, respectively, between the periods 2000-2009 and 2010-2019. However, in the Wet Andes, Hock and Huss (2018) did not estimate any changes in glacier runoff on the western

side of the Andes (Biobio catchment), and instead found a decrease (Río Negro catchment) and an increase (Río Santa Cruz catchment) in glacier runoff on the eastern side of the Andes. Our results for this region show an increase in glacier melt by 8% and a decrease in rainfall on glaciers by -3%."

Reply. We have incorporated that the glacier runoff consider the simulated changes in glacier volume and area in L176 of section Data and methods.

Original text: "From these input data, it is possible to obtain annual outputs such as the surface mass balance, glacier volume and area and monthly glacier melt (snow and ice) and rainfall on glaciers (Figure 1)."

Edited text: "From these input data, it is possible to obtain annual simulated processes such as the surface mass balance, glacier volume and area and monthly glacier melt (snow and ice) and rainfall on glaciers (Figure 1). These two processes compose the glacier runoff which considers the glacier volume and area changes."

L440 – at first use, it may be useful to mention what this id refers to since it's clearly not RGI which the inventory stated was used.

Original text:

"id = 6090629570"

Reply: We replaced all "id = xxxxxx" by "catchment id = xxxxx" in the discussion section.

L466 – typo? "considering a largest glacierized area"? "larger" perhaps?

Original text:

"considering a largest glacierized area due to the use of RGI v6.0"

Edited text:

"considering a larger glacierized area due to the use of RGI v6.0"

L464-469 – This suggests that mass balances over two different time periods are being compared? If that's true, this should definitely not be done as the climate forcing may be completely different. Why not model the same time periods (which OGGM can do) and thus be able to do a proper comparison?

Original text:

"In the La Paz catchment, Soruco et al. (2015) evaluated the mass balance of 70 glaciers (1997-2006) and their contribution to the hydrological regime. In the present study, we simulated a less negative mass balance (-0.56 ± 0.19 m w.e. yr-1 vs. -1 m w.e. yr-1) considering a largest

glacierized area due to the use of RGI v6.0 (with 14.1 km2 in comparison to 8.3 km2). Our estimation of the mean annual glacier runoff (22%) is larger than the previous estimation close to 15% (Soruco et al., 2015). This may be due to the fact that we have considered a warmer 2010-2019 period than the one observed in Soruco et al. (2015)"

Reply: We aim to compare our simulations with local studies to ensure that the orders of magnitude in our simulations are realistic. However, we are not attempting to replicate tighter comparisons as this would considerably lengthen the text. Besides the mentioned period of analysis and climatic forcing, there exist other significant differences between both studies. For instance, the utilization of glacial contours from various glacier inventories and distinct methodologies to estimate glacier runoff. Within the text paragraph, we discuss the utilization of different periods, data, and methods within a regional context. Consequently, the simulation in La Paz is expected to yield results within the order of magnitude previously reported, rather than values estimated for the Dry Andes or Wet Andes.

*L478 – "is" close to?*

Original text:

"Our mean annual glacier contribution estimation close to 15%"

Edited text:

"Our mean annual glacier contribution estimation is close to 15%"

*L479-L481 – sentence does not make sense. Please clarify/rephrase what is being stated.*

Original text:

"It is difficult to compare our results given here with previous studies in the Maipo and La Paz catchments, as any comparison is limited by the use of different input data and models, as well as spatial resolution, time step and calibration processes"

Edited text:

"However, this comparison between our results and previous studies in the Maipo and La Paz catchments is limited due to the utilization of different inputs, spatial resolutions, time steps, and workflow in the simulated processes where some processes as mass balance in all glaciers was not done."

*L492 – temperature index "model" …*

Original text:

"using a temperature index with higher mean values in the Tropical Andes"

Edited text:

"using a temperature-index model with higher mean values in the Tropical Andes"

*L497 – how were all of these differences taken into account? The present study appears to only perform a single set of simulations with one set of model parameters for each glacier. Am I missing something?*

Original text:

"However, these studies considered different scales, both spatially (from stakes to a catchment scale) and temporally (from hourly to monthly), as well as different in situ and fixed (literature) melt factor values for the snow and ice temperature index. Taking these differences into account, we found a regional pattern for the melt factor using the same methodology at a monthly time step".

Edited text:

"However, these studies considered different scales, both spatially (from stakes to a catchment scale) and temporally (from hourly to monthly), as well as different melt factor values for the snow and ice. Here, we have identified a similar regional pattern for the melt factor as the one previously reported, but with an application of a consistent methodology.

*L502 – higher "melt" factor values?*

Original text:

"Dry Andes imply higher factor values to reach the calibrated mass loss in the few months in which the temperatures exceed 0°C."

Edited text:

"Dry Andes imply higher melt factor values to reach the calibrated mass loss in the few months in which the temperatures exceed 0°C."

*L502 & L512-514 – The problem being described seems identical to the overparameterization problem associated with glacier evolution models (e.g., Rounce et al. 2020; JoG). Hence, it might be worth mentioning that this might be what is occurring, i.e., changes in temperature (essentially a temperature bias correction performed using in-situ measurements) are being compensated by changes in the degree-day factor, and thus caution should be taken to avoid overinterpreting calibrated model parameters (especially given that they'll compensate for other factors not accounted for in the model such as changing surface conditions, avalanching, etc. as*

*mentioned in the limitations section). Changing the melt temperature threshold and the rain/snow temperature thresholds by 2 degrees for the Tropical Andes compared to the Dry/Wet Andes.*

*(Table 1) is essentially the exact same thing as using a temperature bias value of 2 degrees (i.e., an additive value that adjusts the temperature by 2 degrees).*

Original text:

L502

"The lowest mean temperatures estimated in the Dry Andes imply higher factor values to reach the calibrated mass loss in the few months in which the temperatures exceed 0°C. The opposite can be observed in the Wet Andes, where low factor values are associated with a greater number of months with temperatures exceeding 0°C"

L511-514

"These differences found in the corrected TerraClimate limit the capacity of the ice/snow melting module to accurately simulate the months in which melting can occur. To account for this, the values of the thresholds used for the melting onset and for the solid/liquid precipitation phase have been adjusted."

Considering the next sentences proposed by Rounce et al. (2020)

> *"One important question to consider is how non-identifiability affects projections of glacier mass change and runoff. Since the model parameters are non-identifiable, the joint posterior distribution will contain different combinations of model parameters that result in equal (or near equal) values of the mass balance. For example, consider two viable sets of model parameters that cause the mass balance to agree with the observation: the first is a wetter and warmer set, i.e., a high precipitation factor compensated by a high temperature bias, and the second is a dryer and cooler set, i.e., a low precipitation factor compensated by a low temperature bias. Present-day glacier mass change will be the same and projections may also be similar, although there may be minor differences that are caused by how the glacier hypsometry impacts the glacier retreat. Conversely, the implications for glacier runoff are likely to be significant both for present-day and future simulations. The wetter and warmer set will generate more precipitation and melt resulting in more glacier runoff, while the dryer and cooler set will result in substantially less glacier runoff.*

> *Similarly, non-identifiability is important to consider for studies that have used glacier models to infer biases in the temperature and precipitation data (e.g., Immerzeel and others, 2015). If the parameters in the model are non-identifiable, then caution must be used in interpreting the results."*

We edited the L511 to 514 adding

"These differences found in the corrected TerraClimate data limit the capacity of the ice/snow melting module to accurately simulate the months in which melting can occur. To account for this, the values of the thresholds used for the melting onset and for the solid/liquid precipitation phase have been adjusted and are described in the limitations (2)."

Incorporating the proposed discussion from L533 when limitations (2) end.

According to Rounce et al. (2020), similar results of glacier surface mass balance could be due to different combinations of model parameters. For instance, a wetter (or dryer) and warmer (or colder) parameter set—where high (or low) precipitation factors are compensated by high (or low) temperature biases—can lead to similar recent glacier mass changes and projections. Conversely, the implications for glacier runoff are likely to be significant for both recent and future simulations. In a wetter (or dryer) and warmer (or colder) scenario, there would be increased (or decreased) precipitation and melt, resulting in larger (or smaller) glacier runoff. To address this, we obtained realistic values for precipitation and temperature based on in-situ spatially distributed measurements and on our field experiences on monitored Andean glaciers. Furthermore, our evaluation of simulations in the three selected catchments enabled us to estimate glacier runoff amounts in the same order of magnitude as previous reports. However, caution must be exercised when using the calibrated melt factors estimated in the Tropical Andes. This is because the temperature in this region was overestimated by an average of 2.4°C, leading to consequences in the calibrated melt factor values. These values should be lower than those estimated here (see Figure 3).

*L530 – how did the geodetic mass balance define the maximum melting per glacier? Again, this may stem from my misunderstanding of what's actually being calibrated and how the calibration was performed.*

Original text:

"Because of the monthly temperature variability, the upper threshold defines the melting onset and determines the number of months in which it occurs. Meanwhile, the geodetic mass balance defined the maximum melting per glacier in a given period"

Edited text:

"Due to the monthly variability in temperature, the melting temperature threshold establishes the onset of melting and influences the number of months in which it occurs. On the other hand, the geodetic mass balance defines the accumulated gain or loss per glacier over the calibration period, which in this case spans 20 years."

*L532 – what does a "true seasonal melting distribution" mean? Is this the observations?*

Original text:

"Based on our evaluation of the corrected TerraClimate temperature and simulated mass balance, we found a true seasonal melting distribution, associated with a mean underestimated mass balance of 185 mm w.e. yr-1 that was highly correlated with the in situ data (r = 0.7)."

Edited text:

"Based on our evaluation of the corrected TerraClimate temperature and simulated mass balance, we correctly reproduce the seasonal melt distribution, associated with a mean underestimated overall annual mass balance of 185 mm w.e. yr-1 which however is highly correlated with the in situ data (r = 0.7)."

*L551-554 – Are these percentage increase in the Tropical Andes and Dry Andes referring to the same Inner Tropic and Dry Andes 1 zones or is this statement meant to refer to different zones? Please clarify as this is unclear.*

Original text:

"The glacier runoff response to this glacier reduction has the largest percentage increase in the Tropical Andes and Dry Andes. Despite this, the largest percentage increase of glacier runoff (> 62%) estimated in the Inner Tropic and Dry Andes 1 zones corresponds to the lowest absolute glacier runoff amounts across the Andes."

Edited text:

[revised manuscript text omitted]

2. Figure 1, for example, doesn't have an arrow going from the corrected data to forcing the simulations. The caption also makes it sound like these are two separate workflows (one input data to run the model and another to perform a correction); however, it's unclear to me how these are done separately if the mass balance model is then calibrated after these are performed?

Reply: We have edited the figure and its description. Please, go to check the answer to the comment L214.

3. More details on how glacier runoff is defined (e.g., fixed vs. moving gauge) and what glacier contribution to runoff means (is this the ratio of the glacier melt to the glacier melt plus precipitation just at the outlet of the glaciers?) are needed. This would help provide context to interpret the results.

In lines 67-68 the original sentences

"Here, using OGGM, we estimate the glacier changes (area and volume) and the consecutive hydrological responses (from glacier melt [ice melt and snow melt] and rainfall on glaciers) for 786 catchments across the Andes (11°N-55°S)"

In relation to this comment, the glacier runoff represents the amount of water coming out from the glacier (snow and ice melt + liquid precip on the glacier). Because of that, we are not simulating the adjacent rock area at a higher elevation to the glacier front and the rock area from the glacier retreat.

The new sentences is

"Here, using OGGM, we estimate the glacier changes (area and volume) and the consecutive hydrological responses called glacier runoff (which is composed of glacier melt [ice melt and

snow melt] and rainfall on glaciers coming from the glacier) for 786 catchments across the Andes (11°N-55°S)"

The meaning of "glacier runoff contribution" was incorporated in the comment related to the L44

4. The novelty of the study is unclear. For example, the conclusion that most glacierized catchments are losing glacier mass is already known from the observations used to calibrate the model. The third conclusion is that the results are consistent with previous studies. It would be useful for the novel aspects of this study to be included.

We edited the section conclusion removing the first and editing the third conclusion

Original text:

"In this study, we present a detailed quantification of the glacio-hydrological evolution across the Andes (11°N-55°S) over the period 2000-2019 using OGGM. Our simulations rely on a glacier-by-glacier calibration of the changes in glacier volume. Simulations cover 36% (11,282 km$^2$) of their glacierized surface area across the Andes where 50% of the area corresponds to the Patagonian icefields and Cordillera Darwin that were not simulated due to specific processes such as calving and which are not accounted for in the version of glaciological model used here. In addition, we used corrected climate forcing and evaluated our simulation results at both the glacier and catchment scale using in situ observations, which are uncommon practices in regional simulations. From our results we can highlight the following:

- 93% of the studied glacierized catchments show a decrease in glacier area between the periods 2000-2009 and 2010-2019, displaying a high coherence with previous reports based, in particular, on glaciers in the Tropical Andes (Rabatel et al., 2012; Seehaus et al., 2020), Wet Andes (Rabassa 2010; Ruiz et al., 2017) and Dry Andes (Rabatel et al., 2011; Malmros et al., 2016; Farías-Barahona et at., 2020).
- The glacier runoff response to this glacier reduction has the largest percentage increase in the Tropical Andes and Dry Andes. Despite this, the largest percentage increase of glacier runoff (> 62%) estimated in the Inner Tropic and Dry Andes 1 zones corresponds to the lowest absolute glacier runoff amounts across the Andes.
- The three selected catchments, located in contrasted climatic zones, are used to evaluate the simulations. They display consistent results with previous studies and in situ observations. The larger glacier contributions to the catchment water flows are quantified for the Baker (43%) and Maipo (36%) catchments during the summer season (January-March). On the contrary, the larger glacier contribution to the La Paz catchment (45%) was estimated during the transition season (September to November).

Lastly, our results help to improve knowledge about the hydrological responses of glaciers in a large part of the Andes through the correction of climate data, the use of the same input data and the same simulation processes as well as a strong glacier calibration applied to the glaciers. The implementation of this calibrated and evaluated model in the historical period is a prerequisite for simulating the future evolution of the Andean glaciers."

Edited text:

"In this study, we present a detailed quantification of the glacio-hydrological evolution across the Andes (11°N-55°S) over the period 2000-2019 using OGGM. Our simulations rely on a glacier-by-glacier calibration of the changes in glacier volume. Simulations cover 36% (11,282 km$^2$) of the glacierized surface area across the Andes where 50% of the total area corresponds to the Patagonian icefields and Cordillera Darwin that were not simulated due to specific processes such as calving and which are not accounted for in the version of glaciological model used here. The simulations were performed for the first time employing the same methodological approach, and a corrected climate forcing and parameter calibration at the glaciological zone scale throughout the Andes. Evaluation of our simulation outputs spanned both glacier-specific and catchment-scale assessments, integrating in situ observations-an unconventional approach within regional simulations. 
[revised manuscript text omitted]

---

## Author Comment (AC2)

**Review of "Hydrological response of Andean catchments to recent glacier mass loss" by Caro et al.**

Dear reviewer, in this document we present our replies (in purple) together with the changes we made to answer your comments. For each of your comments (in *Italics-black*), the original draft text is written in red color, whereas our proposed changes to the original draft text and comments are in blue color.

*This study examines the hydrological response of glaciers in 786 catchments across the Andes in the period 2000-2019 by integrating meteorological data and OGGM. Similar to the comment of the Referee #1, I wonder what is the research gap / justification / novelty of this study considering the availability of observations of glacier mass balance across the Andes (especially the study published by Dussaillant et al. (2019) in NATGEO for the 2000 – 2018 period)? Talking about the hydrological response and referring to Huss and Hock (2018) NCC paper, I would expect bit more elaboration of the peak water timing in different zones of the Andes. Or is the main goal the calibration and performance evaluation of the OGGM model? If so, the study should be re-framed and re-structured in my opinion.*

Reply: Dear reviewer, we sincerely value your feedback and have implemented proposed changes to the article under review. Our primary goal was to analyze the recent changes in the Andean glacier runoff while refining the parameter calibration for melting in OGGM. Additionally, we utilized this calibrated and rigorously evaluated model for a subsequent article, simulating glacier runoff projections across the Andes throughout the 21st century with particular attention to the peak water. We anticipate submitting this second article within the first month of 2024.

As per your suggestions, we have meticulously edited the current article, with a concentrated effort on improving the abstract, introduction, and conclusion sections.

*L19-20: what is the meaning of these %?*

Original text:

"Our results show that the glacier volume (-8.3%) and surface 20 area (-2.2%) are reduced in 93% of the catchments between the periods 2000-2009 and 2010-2019."

Edited text:

"Our results at the Andes scale show that the glacier volume and surface area were reduced by 8.3% and 2.2%, respectively, between the periods 2000-2009 and 2010-2019."

*L36: but the shrinkage did not start in late 1970s, please reformulate*

Original text:

" They have been affected by a continuous shrinkage since the late 1970s, which has intensified during the last two decades (Rabatel et al., 2013; Dussaillant et al., 2019; Masiokas et al., 2020)."

Edited text:

"Continuous glacier shrinkage has been detected since the late 1970s, with intensification observed over the past two decades (Rabatel et al., 2013; Dussaillant et al., 2019; Masiokas et al., 2020)."

*L73-75: please consider deleting*

Original text:

"Section 2 presents the data and methods. In Section 3, we describe the glacier changes and hydrological responses at the glaciological zone and catchment scales across the Andes. In Section 4, we discuss our results and the main steps forward compared to previous research."

Reply: We appreciate your comment; however, this paragraph introduces the main sections to the reader.

*L88: surprisingly, there are no meteorological stations included for 11°N to 9°S where there are different climatological conditions compared to the rest of the study region; please comment on how this gap can impact your analysis especially in the IT zone*

Original text:

[revised manuscript text omitted]

*Fig. 1: please check the completeness of your workflow (e.g. the 3 catchments studies in detail); please also consider linking individual components of your workflow to the sections of the manuscript;*

Original text : "Workflow per glacier simulation using OGGM between 2000 and 2019."

Reply: In Figure 1, we present the workflow of simulation performed in each glacier. We will cite this figure in each step presented in the workflow.

Edited text: "Workflow per simulated glacier using OGGM between 2000 and 2019."

*L223: to what elevation are these numbers referring to?*

Original text:

"The various glaciological regions show significant climatic differences, with contrasting extreme values between the Tropical Andes and Wet Andes in terms of mean annual precipitation (939 ± 261 mm yr-1 and 3751 ± 1860 mm yr-1, respectively) and mean annual temperature between the Dry Andes and Tropical Andes (-3.7 ± 1.4°C and 1.3 ± 0.8°C, respectively)."

Reply: All climate values are related to the mean elevation of glaciers. Considering your question, we edited the section 2.1.3 from L128 to 129

Original text :

" The glacier extent in the RGI v6.0 is representative of the early 2000s."

Edited text:

"The glacier extent in the RGI v6.0 is representative of the early 2000s. The analysis by catchment and glaciological region is related to the locations and elevation of these glaciers."

*L245: please consider displaying a metric quantifying the fit between observed and corrected data in Tab. S2*

Reply: Thank you for your observation. We incorporated the bias between the corrected temperature of TerraClimate and the observations at each meteorological station. These values are equal to the bias presented in Figure S2.

*Fig. 2: please consider incorporating also relative changes in this figure*

Reply: This figure contains several maps and graphs. These show the absolute values, allowing you to directly know the loss rates of volume or area per year.

*Fig. 3: maybe a boxplot everyone can read without additional explanation could work here?*

Reply: We appreciate your comments and agree, however, the boxplot does not show the average or mode.

*L318-320: the comparison of absolute numbers (m3/s) doesn't tell a lot and the comparison is meaningless since these regions don't have comparable glacier coverage; please check here and in other parts of the manuscript that the number you refer to can be compared across the zones / regions*

Original text :

"In addition, the mean annual rainfall on glaciers across the Andes is 387 m3/s for the period 2000-2019. The Wet Andes has the largest amount of annual rainfall (372.7 m3/s), followed by the Tropical Andes (10.5 m3/s) and Dry Andes (4.2 m3 320 /s) with the lowest contribution of rainfall."

Reply: One of the key messages we aim to convey here is the recognition that certain regions receive more water from rainfall on glacier surfaces than others. Specifically concerning rain on glacier surfaces, as of the current article's writing, there is no definitive understanding of the proportion of rainfall occurring during the key seasons in the Tropical, Dry, or Wet Andes. Limited estimates have been made for a handful of glaciers and a few catchments.

*L333: since there is no catchment studied north from La Paz, I wonder why not to use Peruvian Río Santa catchment for which similar data are available and you refer to it in introduction?*

Reply: Indeed, this could have been an option, but we do not want to multiply the examples and we chose 3 (one in each of the "main" climatic zones). La Paz was preferred as the glaciers could be relevant to 2,700,000 people in La Paz and El Alto cities.

*L365-377: this part is more about the climate (changes) and doesn't correspond with the section title; please consider moving*

Reply: This article, though comprehensive, does not primarily aim to quantify variations in climate. Nevertheless, recognizing climate as the principal driver of glacier changes, we consider pertinent to incorporate this aspect. To underscore the significance of climate in influencing glacier dynamics, we believe it is fitting to provide a brief description on the subject. We would appreciate your consideration of these lines as a brief introduction, as consigning these results to supplementary materials might not fully convey their significance.

*Tab. 3: please show and compare simulated and observed values*

Reply: This table presents the simulation results on the scale of the three catchments. The comparison between simulations and observations was conducted at specific locations, and the results are presented in tables and figures in the supplementary materials.

*L418-419: please delete*

Original text:

"In this section we will discuss the relevance of the results obtained at the regional- to glacier-scale across the Andes. We will also discuss the main methodological advantages and limitations of the simulations."

Reply: Thank you. We will delete it.

*L453: please make sure that these studies appear in the list of references*

Original text:

"In the Dry Andes, this correlation was high with precipitation (r = 0.8 ± 0.1) and in the Wet Andes, temperature was correlated with mass 460 balance (r = -0.7 ± 0.1) as previously observed by Caro et al. (2021)."

Reply: We included this reference.

*L459-460: what is the p-value of these correlations?*

Reply: We have edited lines 459-460 to include p-values in the correlations.

Edited text:

"In the Dry Andes, this correlation was high with precipitation (r = 0.8 ± 0.1, p-value < 0.05) and in the Wet Andes, temperature was correlated with mass 460 balance (r = -0.7 ± 0.1,  p-value < 0.05) as previously observed by Caro et al. (2021)."

*L491-495: the main findings of these studies should be summarized in the Introduction and should help you to highlight research gap you are trying to bridge with your study (see my general comment)*

Original text:

"Several reconstructions of the glacier surface mass balance have been performed across the Andes (9-52°S) using a temperature index with higher mean values in the Tropical Andes (0.3-0.5 mm h-1 °C-1), than in the Dry Andes (0.3-0.4 mm h-1 °C-1) and Wet Andes (0.1-0.5 mm h-1 °C-1) (e.g., Fukami & Naruse, 1987; Koisumi and Naruse,1992; Stuefer et al., 1999, 2007; Takeuchi et al., 1995; Rivera, 2004; Sicart et al., 2008; Condom et al., 2011; Caro, 2014; Huss and Hock, 2015; Bravo et al., 2017)."

Reply: Thank you for your feedback. We have relocated this paragraph to the next paragraph in the introduction.

Original text:

[revised manuscript text omitted]

Reply: While we have included certain values within the methodology section, we believe that these values and their detailed descriptions are crucial for facilitating a comprehensive understanding of their significance in the simulated results. Because of that, we called this subsection "Simulation limitations".

*L547: these highlighted points are neither novel nor surprising considering available in situ and remote sensing-based observations of glacier mass balance across the Andes*

Original text:

"93% of the studied glacierized catchments show a decrease in glacier area between the periods 2000-2009 and 2010-2019,"

Reply: Considering all your comments we have reformulated the section Conclusion as

Original text:

"In this study, we present a detailed quantification of the glacio-hydrological evolution across the Andes (11°N-55°S) over the period 2000-2019 using OGGM. Our simulations rely on a glacier-by-glacier calibration of the changes in glacier volume. Simulations cover 36% (11,282 km$^2$) of their glacierized surface area across the Andes where 50% of the area corresponds to the Patagonian icefields and Cordillera Darwin that were not simulated due to specific processes such as calving and which are not accounted for in the version of glaciological model used here. In addition, we used corrected climate forcing and evaluated our simulation results at both the glacier and catchment scale using in situ observations, which are uncommon practices in regional simulations. From our results we can highlight the following:

- 93% of the studied glacierized catchments show a decrease in glacier area between the periods 2000-2009 and 2010-2019, displaying a high coherence with previous reports based, in particular, on glaciers in the Tropical Andes (Rabatel et al., 2012; Seehaus et al., 2020), Wet Andes (Rabassa 2010; Ruiz et al., 2017) and Dry Andes (Rabatel et al., 2011; Malmros et al., 2016; Farías-Barahona et at., 2020).
- The glacier runoff response to this glacier reduction has the largest percentage increase in the Tropical Andes and Dry Andes. Despite this, the largest percentage increase of glacier runoff (> 62%) estimated in

[revised manuscript text omitted]

- - -

*To sum up, I'm convinced this study would benefit from (rather major) revisions regarding its structure, justification and framing in the context of the existing studies.*

Reply: In response to the reviewer's suggestions, we propose to modify the article's structure to enhance the clarity about the primary goal concerning recent changes in glaciers and glacier runoff, or to address corrections and calibration. However, it's important to note that our original intention was aligned with these two goals. Consequently, we have reworked the Abstract, Introduction, and Conclusions sections to get more clear research. The Abstract and Introduction are edited as follows.

Abstract

Original text:

"The impacts of the accelerated glacier retreat in recent decades on runoff changes are still unknown in most Andean catchments, thereby increasing uncertainties in estimating and managing water availability. Here, we used a monthly time step to simulate glacier evolution and related runoff changes for 36% of the glacierized surface area of the Andes (11,282 km$^2$ in 786 catchments, 11°N-55°S) using the Open Global Glacier Model (OGGM) and a corrected and evaluated version of the TerraClimate dataset between 2000 and 2019. The glacier mass balance and volume were calibrated glacier-by-glacier. The simulation results were evaluated with in situ data in three documented catchments and 15 glaciers. Our results show that the glacier volume (-8.3%) and surface area (-2.2%) are reduced in 93% of the catchments between the periods 2000-2009 and 2010-2019. This glacier loss is associated with

changes in climate conditions (precipitation = -9%; temperature = +0.4 ± 0.1°C) inducing an increase in the mean annual glacier melt of 12% (86.5 m$^3$/s) and a decrease in the mean annual rainfall on glaciers of -2% (-7.6 m$^3$/s). We find a regional pattern in the melt factors showing decreasing values from the Tropical Andes toward the Wet Andes. A negative mass balance trend is estimated in the three documented catchments (glacierized surface area > 8%), showing the largest mean glacier contribution during the transition season (September-November) in La Paz (Bolivia) (45%) followed by Baker (Chile) (43%) and Maipo (Chile) (36%) during the summer season (January-March). In addition, our evaluation in the monitored glaciers indicates an underestimation of the mean simulated mass balance by 185 mm w.e. yr$^{-1}$ and a high mean correlation (r = 0.7). We conclude that the large increases in the simulated glacier melt in the Dry Andes (36%) and the Tropical Andes (24%) have helped to improve our knowledge of the hydro-glaciological characteristics at a much wider scale than previous studies, which focused more on a few select catchments in the Andes.".

Edited text:

"The impacts of the accelerated glacier retreat in recent decades on glacier runoff changes are still unknown in most Andean catchments, intensifying uncertainties in estimating water availability. This particularly affects the Outer tropics and Dry Andes, heavily impacted by prolonged droughts. Current global estimates overlook climatic and morphometric disparities among Andean glaciers, which significantly influence simulation parameters. Meanwhile, local studies have used different approaches to know glacier runoff in a few catchments. Enhanced accuracy in 21st-century glacier runoff projections hinges on corrected historical climate 
[revised manuscript text omitted]

---

## Referee Report (RR1)

**Review of "Hydrological response of Andean catchments to recent glacier mass loss"**
**by Caro et al.**

This study investigates the changes in glacier mass, area, and runoff for different glaciated catchments in the Southern Andes from 2000-2019. The study uses the Open Global Glacier Model, calibrated with geodetic mass balance data from 2000-2019, and forced by a bias-corrected climate dataset. The focus of the study is on all land-terminating glaciers (i.e., lake- and marine-terminating glaciers are excluded) and specific attention is given to how the changes in climate over 2000-2009 vs. 2010-2019 affect the glacier runoff across the different catchments, which notably span different climatologies. The main conclusions are that most glaciers are losing mass leading to increases in runoff in the Tropical Andes and Dry Andes. Furthermore, results are consistent with previous studies and the glacier contribution to runoff is highest for some catchments in the summer and others in the transition season prior to summer.

This is the second time I'm reviewing this article and I'm pleased to say the manuscript is greatly improved! Excellent job. The methods are now incredibly detailed and thus easy to understand what was done and what the advances are. Specifically, the major advance/novelty is the use of a bias-corrected climate dataset and the in-situ glaciological observations. Hence, even though the study primarily confirms existing knowledge (although they do not some key differences and do a nice job framing results with respect to previous studies in the discussion), it is a valuable contribution to the literature.

My comments are primarily related to improving readability as several areas were a bit unclear; however, rephrasing these or being more explicit will resolve these issues easily. Similarly, several areas felt quite repetitive, so sentences/sections could be removed, which would also reduce the length of now a fairly long study. I thus would suggest accepting subject to minor revisions. General and specific comments are described below.

General Comments
Abstract is very detailed. Suggest shortening and highlighting only the key research findings.

Methods are now very detailed. Thanks for this.

Section 4.3 states there is a similar regional pattern; however, this is largely due to overparameterization issues with the model and the assumption of having different temperatures of melt onset for the different regions (L263-264). The authors should mention this overparameterization issue and avoid overinterpreting their "regional patterns". That said, this is well discussed in Section 4.4. I would recommend removing Section 4.3.

Specific Comments
L23 – "emphasize on" consider new word choice.

L24 – repetitive of L20.

L25-27 – likely too detailed for abstract, which is already quite long.

L32-33 – unclear what these values refer to as there are two variables and two periods, so would expect 4 numbers.  Please clarify.

L53-57 – need to provide context of the time period these results are being discussed in.  I assume it's end of century.

L57-58 – "… overlook the diverse climates and morphologies of Andean glaciers" is very vague. Climate data was used, which should account for some diversity of climate.  Unclear if morphologies is referring to glacier types (e.g., land-terminating, marine-terminating) or something different. Please be specific to help with readability.

L84 – suggest removing the word "precisely" as this is a bit misleading for global modeled products.  The sentence appears to be describing that the model can be applied at the glacier scale as opposed to how accurate it can predict changes.

L96-97 – "Whereas…" is not a complete sentence.

L107 – suggest just referencing Figure 1 in parentheses after the previous sentence and deleting this sentence.

L145 – suggest removing results from the methods section.

L240 – "In a second time," I assume this refers to post-processing?  Perhaps "After simulations were completed, …" or this could be deleted.

L277-282 – this feels very repetitive of earlier in the methods.  Given the length of the methods now, I would recommend deleting this.

Table2 and throughout, I highly encourage zones to just be listed as "Dry Andes 1", "Dry Andes 2", etc. to improve readability.  When reading comparisons of DA1 and WA2 (e.g., L297) it becomes very hard to follow.

L323-331 – The first sentence is incredibly hard to follow.  Please rephrase.  I think what is being stated is simply that only 36% of the total glacierized area of the Andes was simulated. This whole paragraph could likely be one sentence and likely belongs in the methods.

L32-333 – same issue as with abstract.  Two time periods and two variables are being reported, yet only two values are shared.  Please clarify.

Figure 2 – "SMB" is currently being used for "simulated mass balance".  It is clear that this is a modeled product, so including "simulated" is unnecessary.  I recommend removing this S.  This also will avoid confusion with the common acronym SMB for surface mass balance.  I'll note that this also is inconsistent with Figure 5 which uses SMB for "specific" mass balance.

L369 – be explicit what the model limitations are.  Is sublimation, which is described as a limitation for the Dry Andes, also a problem for the Tropical Andes?

Figure 5 – specify if the mass balance is cumulative or not.  It appears that it is cumulative.

L666 – the conclusion mentions the "accuracy" that has been improved, but it's unclear what this "enhanced accuracy" is being compared to.  Later in this bullet it mentions compared to global values, but I did not see any estimates of what the error was for global models; hence, how can one state that these are more accurate when it's unknown how accurate the other models are?

References – check that all studies included in the text are included in the references.  Rounce et al. (2020) is cited but not in the references.

---

## Author Response (AR2)

**Review of "Hydrological response of Andean catchments to recent glacier mass loss"**
**by Caro et al.**

Dear reviewer, we appreciate your valuable comments.

In this document, we present our replies (in purple) together with the changes we made to answer your comments. For each of your comments (in *Italics-black*), the text corrected for the 1st round of review is written in red color, whereas our proposed changes to your comments for this second round of revisions are in blue.

**Referee #1**

*This study investigates the changes in glacier mass, area, and runoff for different glaciated catchments in the Southern Andes from 2000-2019. The study uses the Open Global Glacier Model, calibrated with geodetic mass balance data from 2000-2019, and forced by a bias corrected climate dataset. The focus of the study is on all land-terminating glaciers (i.e., lake and marine-terminating glaciers are excluded) and specific attention is given to how the changes in climate over 2000-2009 vs. 2010-2019 affect the glacier runoff across the different catchments, which notably span different climatologies. The main conclusions are that most glaciers are losing mass leading to increases in runoff in the Tropical Andes and Dry Andes. Furthermore, results are consistent with previous studies and the glacier contribution to runoff is highest for some catchments in the summer and others in the transition season prior to summer. This is the second time I'm reviewing this article and I'm pleased to say the manuscript is greatly improved! Excellent job. The methods are now incredibly detailed and thus easy to understand what was done and what the advances are. Specifically, the major advance/novelty is the use of a bias-corrected climate dataset and the in-situ glaciological observations. Hence, even though the study primarily confirms existing knowledge (although they do not some key differences and do a nice job framing results with respect to previous studies in the discussion), it is a valuable contribution to the literature.*

*My comments are primarily related to improving readability as several areas were a bit unclear; however, rephrasing these or being more explicit will resolve these issues easily. Similarly, several areas felt quite repetitive, so sentences/sections could be removed, which would also reduce the length of now a fairly long study. I thus would suggest accepting subject to minor revisions. General and specific comments are described below.*

***General Comments***

*Abstract is very detailed. Suggest shortening and highlighting only the key research findings.*

We edited the abstract. It was shortened from 472 words to 314 words

Text corrected for the 1st round of review:

"Abstract.

[revised manuscript text omitted]

*Methods are now very detailed. Thanks for this.*

*Section 4.3 states there is a similar regional pattern; however, this is largely due to overparameterization issues with the model and the assumption of having different temperatures of melt onset for the different regions (L263-264). The authors should mention this*

*overparameterization issue and avoid overinterpreting their "regional patterns". That said, this is well discussed in Section 4.4. I would recommend removing Section 4.3.*

We agree, we removed Section 4.3

**Specific Comments**

*L23 – "emphasize on" consider new word choice.*
We removed L23
Text corrected for the 1st round of review:
"We also emphasize on climate correction, parameters calibration, and results evaluation within the workflow simulation."

*L24 – repetitive of L20.*
Reply: We deleted L24
Text corrected for the 1st round of review:
"Our homogeneous methodological framework across the Andes considers the diverse glaciological zones in the Andes."

*L25-27 – likely too detailed for abstract, which is already quite long.*
Text corrected for the 1st round of review:
"The atmospheric variables from the TerraClimate product were corrected using in situ measurements, underlining the use of local temperature lapse rates. Meanwhile, the glacier mass balance and volume were calibrated glacier-by-glacier. Furthermore, procedures by glaciological zones allow us to correct mean temperature bias up to 2.1°C and increase the amount of monthly precipitation."
Edited text:
"TerraClimate atmospheric variables were corrected using in situ data, getting a mean temperature bias by up to 2.1°C and enhanced monthly precipitation."

*L32-33 – unclear what these values refer to as there are two variables and two periods, so would expect 4 numbers. Please clarify.*
Text corrected for the 1st round of review:

[revised manuscript text omitted]

*Table2 and throughout, I highly encourage zones to just be listed as "Dry Andes 1", "Dry Andes 2", etc. to improve readability. When reading comparisons of DA1 and WA2 (e.g., L297) it becomes very hard to follow.*

We agree. We edited the new version with your suggestions.

*L323-331 – The first sentence is incredibly hard to follow. Please rephrase. I think what is being stated is simply that only 36% of the total glacierized area of the Andes was simulated. This whole paragraph could likely be one sentence and likely belongs in the methods.*

Text corrected for the 1st round of review:

"The annual mass balance and glacier dynamics per glacier are simulated by considering 36% of the total glacierized surface area across the Andes (11°N-55°S) to obtain the glacier area and glacier volume at an annual time scale, as well as the glacier runoff (glacier melting and rainfall on glaciers) at a monthly time scale. In more details, over 85% of the glacierized surface area in the Dry Andes (18°S-37°S) and 79% in the Tropical Andes (11°N-18°S) is considered, which corresponds to 11% (3,377 km$^2$, in 321 catchments) of the total glacierized area of the Andes. For the Wet Andes (37°S-55°S), 29% of the glacierized surface area in the region is considered, which corresponds to 26% (7,905 km$^2$, in 465 catchments) of the total area in the Andes (see the distribution of the catchments in Figure 2a). The simulated lower glacierized surface area in the Wet Andes results from the filtering out of the numerous calving glaciers found there."

Edited text:

"The 36% of the total glacierized surface area across the Andes (11°N-55°S) are simulated to obtain annual glacier area and glacier volume, as well as the monthly glacier runoff (glacier melting and rainfall on glaciers)."

We moved the other lines to Section 2.1.3 Glacier data in Glacier inventory.

"Overall, 36% of the total glacierized surface area across the Andes is considered. Over 85% of the glacierized surface area in the Dry Andes (18°S-37°S) and 79% in the Tropical Andes (11°N-18°S) are considered, which corresponds to 11% (3,377 km$^2$, in 321 catchments) of the

total glacierized area of the Andes. For the Wet Andes (37°S-55°S), 29% of the glacierized surface area in the region is considered, which corresponds to 26% (7,905 km$^2$, in 465 catchments) of the total glacierized area in the Andes (see the distribution of the catchments in Figure 2a). The simulated glacierized surface area is lower in the Wet Andes due to the filtering out of the numerous calving glaciers found there."

*L32-333 – same issue as with abstract. Two time periods and two variables are being reported, yet only two values are shared. Please clarify.*

Text corrected for the 1st round of review:

"Between the periods 2000-2009 and 2010-2019, the glacier volume and area in the Andean catchments decreases by -8.3% (-59.1 km$^3$) and -2.2% (-245 km$^2$), respectively, associated with a mean annual mass balance of -0.5 ± 0.3 m w.e. yr$^{-1}$ (Figure 2d)."

Edited text:

"Considering mean values for the periods 2000-2009 and 2010-2019, the glacier volume and surface area in the Andean catchments show a decrease by -8.3% (-59.1 km$^3$) and -2.2% (-245 km$^2$), respectively. This corresponds to a mean annual mass balance difference between the two periods of -0.5 ± 0.3 m w.e. yr$^{-1}$ (Figure 2d)."

*Figure 2 – "SMB" is currently being used for "simulated mass balance". It is clear that this is a modeled product, so including "simulated" is unnecessary. I recommend removing this S. This also will avoid confusion with the common acronym SMB for surface mass balance. I'll note that this also is inconsistent with Figure 5 which uses SMB for "specific" mass balance.*

We agree with your comment. We will change "SMB" by "MB" in the figures and along the text.

Text corrected for the 1st round of review:

"The (d) annual simulated mass balances are presented in each glaciological zone"

Edited text:

"The (d) annual specific mass balances are presented in each glaciological zone"

*L369 – be explicit what the model limitations are. Is sublimation, which is described as a limitation for the Dry Andes, also a problem for the Tropical Andes?*

Text corrected for the 1st round of review:

"Model limitations are observed in the Zongo glacier (r = 0.3 and bias = -224 mm w.e. yr$^{-1}$) in the Tropical Andes. In the Dry Andes, no correlation is observed in the three monitored glaciers

(Guanaco, Amarillo and Ortigas 1); this is mainly because sublimation, an ablation process that is not represented in the model, is dominant for these glaciers."

We edited these lines and also we incorporated new references.

Edited text:

"Model limitations are observed on the Zongo glacier (r = 0.3 and bias = -224 mm w.e. yr$^{-1}$) in the Tropical Andes. In the Dry Andes 1, no correlation is observed in the three monitored glaciers (Guanaco, Amarillo and Ortigas 1); this is mainly because sublimation is very high on these glaciers, reaching 81% of the annual ablation (MacDonell et al., 2013). On the other hand, sublimation is lower southward in the Dry Andes 2 with 7% of the annual ablation (Ayala et al., 2017). For the tropical zone, sublimation is close to 13% in Outer Tropics (Sicart et al., 2005) and 5% in Inner Tropics (Favier et al., 2004)."

Ayala, Á., Pellicciotti, F., MacDonell, S., McPhee, J., Burlando, P. Patterns of glacier ablation across North-Central Chile: Identifying the limits of empirical melt models under sublimation-favorable conditions. Water Resources Research, 53(7), 5601– 5625. https://doi.org/10.1002/2016WR020126, 2017.

MacDonell, S., Kinnard, C., Mölg, T., Nicholson, L., Abermann, J. Meteorological drivers of ablation processes on a cold glacier in the semi-arid Andes of Chile. Cryosphere 7:1513–1526. https://doi.org/10.5194/tc-7-1513-2013, 2013.

Favier, V., Wagnon, P., Chazarin, J.-P., Maisincho, L., and Coudrain, A. One-year measurements of surface heat budget on the ablation zone of Antizana glacier 15, Ecuadorian Andes, J. Geophys. Res., 109, D18105, https://doi.org/10.1029/2003JD004359, 2004.

Sicart, J. E., Wagnon, P., and Ribstein, P. Atmospheric controls of heat balance of Zongo Glacier (16°S. Bolivia). J. Geophys. Res. 110:D12106. https://doi.org/10.1029/2004JD005732, 2005.

*Figure 5 – specify if the mass balance is cumulative or not. It appears that it is cumulative.*

Text corrected for the 1st round of review:

"Recent specific mass balance"

In Figure 5, MB is not cumulative.

Edited text:

"Recent annual specific mass balance"

*L666 – the conclusion mentions the "accuracy" that has been improved, but it's unclear what this "enhanced accuracy" is being compared to. Later in this bullet it mentions compared to global values, but I did not see any estimates of what the error was for global models; hence,*

*how can one state that these are more accurate when it's unknown how accurate the other models are?*

We appreciate this comment. You are right, so we edited these lines as follows.

Text corrected for the 1st round of review:

"The correction of temperature and precipitation data, coupled with parameter calibration conducted at the glaciological zone scale, notably enhanced the accuracy of mass balance simulations and glacier runoff estimations."

Edited text:

"The correction of temperature and precipitation data, coupled with parameter calibration conducted at the glaciological zone scale, enabled obtaining  annual estimates of glacier mass balance and runoff closer to what has been measured in glaciers and some Andean catchments."

*References – check that all studies included in the text are included in the references. Rounce et al. (2020) is cited but not in the references.*

We checked the references and included the one by Rounce et al. (2020).

**Referee #2**

*I thank the authors for the revisions they made. While the Introduction section is substantially reworked, it is still bit difficult to understand what actual goal and expected utilization of results - considering existing knowledge about Andean glaciers - are. Instead, the authors describe what they did (second to last paragraph of Intro), but fail to buid the story explaining and justifying why. I appreciate the amount of the work done but I ask the authors to kindly invest additional time to frame, justify and put into context their study appropriately. I recommend minor revisions.*

We appreciate your comments. We edited the Introduction section on the basis of the last comments of Referee 1. Additionally, we edited the Introduction to better articulate our research goals and highlight the potential future applications of our results.

Text corrected for the 1st round of review in red and edited text in blue

[revised manuscript text omitted]